# GTA: Generative Trajectory Augmentation with Guidance for Offline Reinforcement Learning

**Jaewoo Lee**[1] *  **Sujin Yun**[1] *  **Taeyoung Yun**[1]  **Jinkyoo Park**[1][2]

[1]KAIST     [2]Omelet

{jaewoo, yunsj0625, 99yty, jinkyoo.park}@kaist.ac.kr

## Abstract

Offline Reinforcement Learning (Offline RL) presents challenges of learning effective decision-making policies from static datasets without any online interactions. Data augmentation techniques, such as noise injection and data synthesizing, aim to improve Q-function approximation by smoothing the learned state-action region. However, these methods often fall short of directly improving the quality of offline datasets, leading to suboptimal results. In response, we introduce **GTA**, Generative Trajectory Augmentation, a novel generative data augmentation approach designed to enrich offline data by augmenting trajectories to be both high-rewarding and dynamically plausible. GTA applies a diffusion model within the data augmentation framework. GTA partially noises original trajectories and then denoises them with classifier-free guidance via conditioning on amplified return value. Our results show that GTA, as a general data augmentation strategy, enhances the performance of widely used offline RL algorithms across various tasks with unique challenges. Furthermore, we conduct a quality analysis of data augmented by GTA and demonstrate that GTA improves the quality of the data. Our code is available at https://github.com/Jaewoopudding/GTA

## 1   Introduction

Learning a decision-making policy through continual interaction with real environments is challenging when online interaction is costly or risky. Offline Reinforcement Learning [Offline RL, 1] emerges as a solution, focusing on training effective decision-making policies with static dataset gathered through an unknown policy [2]. However, offline data often does not provide enough coverage of state-action space, resulting in extrapolation error addressed as overestimation of Q-value [3, 4]. To mitigate extrapolation error, many previous works rely on adding explicit regularization terms [3–10] and have shown significant progress.

Moving beyond these mainstream methodologies, there exists an underexplored approach, data augmentation methods: traditional data augmentation and generative data augmentation. Traditional augmentation methods [11, 12] inject minimal noise to the state, preserving environment dynamics. Generative data augmentation [13, 14] builds a data synthesizer with a generative model to upsample offline data. Employing generative data augmentation approaches broadens the support of the data, thereby improving Q-function approximation [12].

While previously proposed data augmentation methods enhance the performance of offline RL, they still have limitations. As illustrated in Figure 1, traditional augmentation methods have difficulties in discovering novel states or actions beyond the existing offline data, limiting their role only to smoothing small local regions of the observed state. In the case of generative data augmentation, the reward distribution of generated data is constrained by the support of offline data, resulting in the

---

*Equal contribution authors.

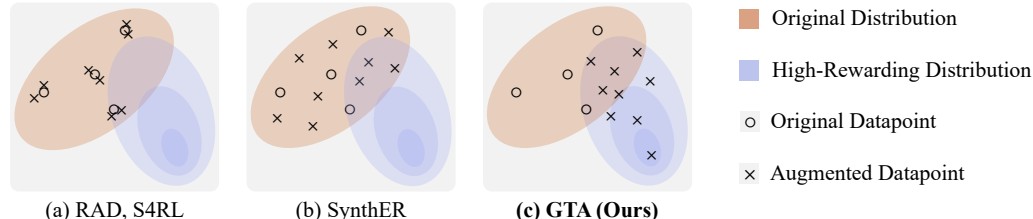

|   | | |
|---|---|---|
| (a) RAD, S4RL | (b) SynthER | **(c) GTA (Ours)** |

Figure 1: Comparison of noise injection [11, 12], generative data augmentation [13] and GTA.

generation of suboptimal data. These issues make the existing methods not align with the objective of offline RL, which aims to learn the most effective decision-making policy from the static dataset.

We introduce Generative Trajectory Augmentation (GTA), a novel approach that applies the conditional diffusion model to data augmentation, aiming to address the aforementioned limitations. GTA is designed to generate novel and high-rewarding trajectories while minimizing degradation of dynamic plausibility. Our approach consists of three main stages: (1) Training a conditional diffusion model that generates *trajectory-level data*, (2) Augmenting the offline data via *partial noising with diffusion forward process* and *denoising with amplified return guidance*, and (3) Training any offline RL algorithm with augmented data.

We train a conditional diffusion model that approximates the conditional distribution of trajectory given its return. Once the diffusion model is trained, we sample trajectories from the offline dataset and *partially noise* the trajectory with the diffusion forward process. Then, we denoise the noised trajectory with *amplified return guidance*, directing trajectories to the high-rewarding region. To this end, we can orthogonally integrate data augmented by GTA into any offline RL algorithm without any modification.

**Our contributions**. In this paper, we introduce GTA, a novel data augmentation framework that utilizes a conditional diffusion model to generate high-rewarding, novel, and dynamically plausible data. Through extensive experiments on commonly studied benchmarks [15, 16], we demonstrate that GTA significantly improves performance across various tasks with unique challenges, such as sparse reward tasks and high-dimensional robotics tasks. We thoroughly examine the impact of core design components of GTA, *trajectory generation*, *partial noising with diffusion forward process* and *denoising with amplified return guidance*. Furthermore, we assess the GTA augmented dataset with data quality metrics, proving its alignment with the objective of offline RL. These findings underscore the capability of GTA to efficiently augment trajectories, resulting in high-quality samples and improving the performance of offline RL algorithms.

## 2 Related Works

**Data Augmentation for Reinforcement Learning**. In Reinforcement Learning (RL), data augmentation enhances sample efficiency and improves Q-function approximation. In pixel-based RL, methods like CURL [17], RAD [11], and DrQ [18] have leveraged image augmentations such as cropping and translation to address sample-efficiency of RL. For the proprioceptive observations, S4RL [12] introduces variations into states by adding small-scale noise or adversarial gradients of Q-function under the similarity prior, that similar states yield similar rewards. AWM [19] improves robustness of augmented data by applying simple transformations to the learned dynamics model, enabling zero-shot generalization to unseen dynamics without requiring multiple test-time rollouts.

Recent strides in generative models have led to their adoption in generative data augmentation, as seen in SynthER [13], MTDiff-S [14], and PGD [20]. Among these, GTA, MTDiff-S, PGD, and SynthER all generate synthetic data using diffusion models but with distinct approaches. While GTA, MTDiff-S, and PGD generate at the trajectory level, MTDiff-S is designed for multi-task settings, focusing on generating trajectories for unseen tasks. PGD generates synthetic trajectory with classifier guidance of policy, similar to model-based offline RL. GTA, on the other hand, focuses on generating high-rewarding trajectories using return guidance. Focusing on single tasks, GTA and SynthER differ in what they generate. SynthER generates individual transitions, in contrast to the trajectory-level generation of GTA. This makes GTA compatible with any kind of offline RL model. Additionally, GTA initializes augmentation from the original trajectory, utilizing a conditional diffusion model, while SynthER starts augmentation from the Gaussian noise without conditional guidance.

**Diffusion Models for Offline Reinforcement Learning**. In focus on the diffusion planners, Diffuser [21] devises the RL problem as a trajectory-level generation with cumulative reward classifier guidance. Following this, the Decision Diffuser [22] replaces the classifier guidance of the Diffuser with classifier-free guidance [23]. Adaptdiffuser [24] alternates between generating trajectories with diverse reward functions and training the model using self-generated trajectories. However, diffusion planners require extensive time to sample actions, making their practical application challenging. GTA shifts the computational burden of the diffusion model from the decision-making step to the data-preparation step. This transfer allows GTA to leverage the advantages of diffusion models while avoiding extensive time costs during decision-making.

**Model Based Offline Reinforcement Learning**. Model-based Offline RL focuses on learning environment dynamics and reward functions from data, creating simulated trajectories to train both the critic and policy [25–28]. While both GTA and model-based offline RL generate synthetic trajectories, in the case of GTA, a key difference is that data generation and policy learning are separated and not done in an alternative cycle. This separation ensures that the quality of generated data is not affected by the training progress of critics or policies. It also mitigates the accumulating errors associated with single-step dynamics rollouts [22].

## 3 Preliminaries

### 3.1 Offline Reinforcement Learning

Reinforcement Learning (RL) is modeled with the Markov decision process (MDP) described by the tuple $(\mathcal{S}, \mathcal{A}, \mathcal{T}, \mathcal{R}, \gamma)$, consisting of state space $\mathcal{S}$, action space $\mathcal{A}$, transition function $\mathcal{T} : \mathcal{S} \times \mathcal{A} \to \mathcal{S}$, reward function $\mathcal{R} : \mathcal{S} \times \mathcal{A} \times \mathcal{S} \to \mathbb{R}$, and discount factor of future reward $\gamma \in [0, 1)$ [29]. At each timestep $t$, the agent selects an action $a_t$ according to the policy $\pi$ given the state $s_t$. Consequently, the agent receives a reward $r_t$ for the action $a_t$ taken in the state $s_t$, leading to the next state $s_{t+1}$. The goal of RL is to learn policy $\pi^*$, which maximizes expected discounted return, $J(\pi) = \mathbb{E}_\pi \left[ \sum_{t=0}^\infty \gamma^t r_t \right]$.

In the offline RL setting, we can only have access to the fixed dataset $\mathcal{D}$, which has been collected using unknown behavior policy $\pi_\beta$. With insufficient and suboptimal offline data, offline RL aims to learn effective decision-making policy that surpasses the behavior policy.

### 3.2 Diffusion Models

**Score-based diffusion models**. Diffusion models are a family of generative models that approximate the data distribution $p(\boldsymbol{x})$ with $p_\theta(\boldsymbol{x})$. When considering the data distribution as $p(\boldsymbol{x})$ and the standard deviation of noise as $\sigma$, then the distribution of data with added noise is denoted as $p(\boldsymbol{x}; \sigma)$. The diffusion reverse process involves sequentially denoising from noise $\boldsymbol{x}^K$ that randomly sampled from a Gaussian distribution $\mathcal{N}(0, \sigma_{\max}^2 I)$, following a noise level sequence $\sigma_K = \sigma_{max} > \sigma_{K-1} > \cdots > \sigma_0 = 0$. Consequently, the endpoint $\boldsymbol{x}^0$ of this process aligns with the original data distribution.

Considering the probability flow ordinary differential equations (probability flow ODE), noise is continuously added to data during the forward process and reduced in the reverse process. By scheduling the noise level at time $k$, represented as $\sigma(k)$, the reverse probability flow ODE is formulated as follows [30]:

$$\mathrm{d}\boldsymbol{x} = -\dot{\sigma}(k)\sigma(k)\nabla_{\boldsymbol{x}} \log p(\boldsymbol{x}; \sigma(k))\mathrm{d}k \tag{1}$$

where $\nabla_{\boldsymbol{x}} \log p(\boldsymbol{x}; \sigma(k))$ denotes the score function, signifies the direction towards the data for a given noise level, and the dot represents the time derivative. The score function $\nabla_{\boldsymbol{x}} \log p(\boldsymbol{x}; \sigma(k))$ is trained via denoising score matching. The denoising score matching loss for given denoiser $D_\theta(\boldsymbol{x}; \sigma)$ is given by

$$\mathcal{L}(D_\theta; \sigma) = \mathbb{E}_{\boldsymbol{x} \sim p, \epsilon \sim \mathcal{N}(0, \sigma^2 I)} \|D_\theta(\boldsymbol{x} + \epsilon; \sigma) - \boldsymbol{x}\|_2^2. \tag{2}$$

When the denoiser $D_\theta(\boldsymbol{x}; \sigma)$ is optimally trained, the score is calculated as

$$\nabla_{\boldsymbol{x}} \log p(\boldsymbol{x}; \sigma) = (D_\theta(\boldsymbol{x}; \sigma) - \boldsymbol{x})/\sigma^2. \tag{3}$$

We sample data via solving Equation (1) with the learned denoising network.

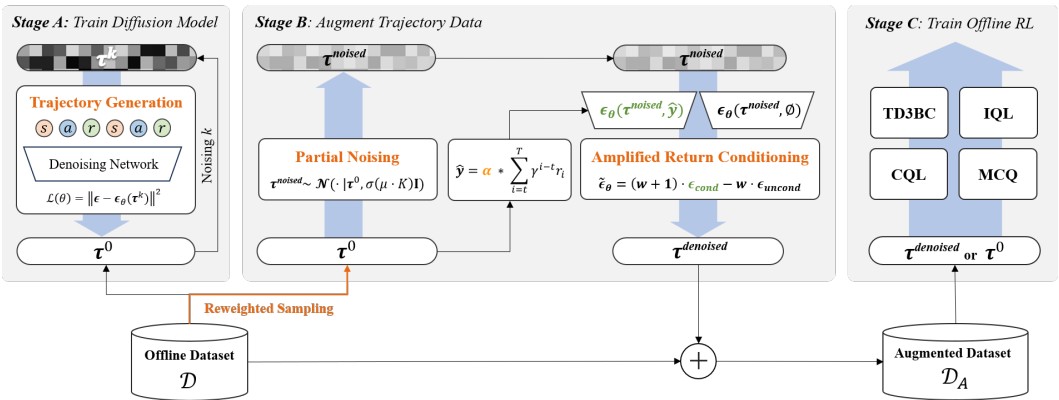

Figure 2: Overall framework of the GTA comprises 3 major stages. In the first stage, we train a conditional diffusion model designed for generating trajectories. Following this, We perturb the original trajectory and subsequently denoise it using the trained diffusion model, conditioned by amplified return. Lastly, we employ the augmented dataset to train various offline RL algorithms.

**Conditional score-based diffusion model**. In the domain of conditional diffusion models, two primary strategies are recognized: classifier guidance [31] and classifier-free guidance [23]. Classifier free guidance sets its guidance distribution $\tilde{p}_\theta$ as $\tilde{p}_\theta(\boldsymbol{x}|y) \propto p_\theta(\boldsymbol{x}|y) \cdot p_\theta(y|\boldsymbol{x})^w$. Subsequently, using the implicit classifier $p_\theta(y|\boldsymbol{x}) \propto p_\theta(\boldsymbol{x}|y)/p_\theta(\boldsymbol{x})$ and the equivalence relationship between score matching and denoising process, described as $\nabla_{\boldsymbol{x}} \log p_\theta(\boldsymbol{x}|y) \propto \epsilon_\theta(\boldsymbol{x}, y)$ [32, 23], the classifier free guidance score $\tilde{\epsilon}_\theta$ is formed as follows:

$$\tilde{\epsilon}_\theta(\boldsymbol{x}^k|y) = (w+1) \cdot \epsilon_\theta(\boldsymbol{x}^k, y) - w \cdot \epsilon_\theta(\boldsymbol{x}^k, \varnothing) \tag{4}$$

where $w$ controls the strength of the guidance. The training objective of the classifier free guidance is to concurrently train the conditional score function and the unconditional score function as follows:

$$\mathcal{L}(\theta) = \mathbb{E}_{k,\epsilon,\boldsymbol{x},\beta} \left[ \left\| \epsilon - \epsilon_\theta \left( \boldsymbol{x}^k, (1-\beta)y + \beta\varnothing \right) \right\|^2 \right] \tag{5}$$

where $y$ is a condition and $\beta \sim \text{Bern}(\lambda)$ is a binary variable with dropout rate $\lambda$ of condition $y$.

## 4 Method

In this section, we introduce **GTA**, **G**enerative **T**rajectory **A**ugmentation, which leverages the conditional diffusion model to generate high-rewarding, novel, and dynamically plausible trajectories for augmentation. As shown in Figure 2, our method is divided into three stages: (1) Train a conditional diffusion model at the trajectory level. (2) Augment trajectories using a *partial noising and denoising framework* with *amplified return guidance*. (3) Train any offline RL algorithm with the augmented dataset. We will discuss how each component works and their roles in effective data augmentation.

### 4.1 Stage A: Train Diffusion Model

**Trajectory-level generation**. The diffusion model for GTA is designed to generate the subtrajectory $\boldsymbol{\tau}$. This subtrajectory is a consecutive transition sequence of the state, action, and reward sampled from a trajectory $(s_1, a_1, r_1, ..., s_T, a_T, r_T)$. We represent subtrajectory $\boldsymbol{\tau}$ with horizon $H$ as follows:

$$\boldsymbol{\tau} = \begin{bmatrix} s_t & s_{t+1} & \cdots & s_{t+H-1} \\ a_t & a_{t+1} & \cdots & a_{t+H-1} \\ r_t & r_{t+1} & \cdots & r_{t+H-1} \end{bmatrix} \tag{6}$$

We train a conditional diffusion model to approximate the conditional distribution $p(\boldsymbol{\tau}|y(\boldsymbol{\tau}))$ with offline dataset, where the condition $y(\boldsymbol{\tau}) = \sum_{i=t}^{T} \gamma^{i-t} r_i$ denotes the sum of the discounted return. The parameter $\theta$ is updated to maximize the expected log-likelihood of the data:

$$\theta^* \leftarrow \arg\max_\theta \mathbb{E}_{\boldsymbol{\tau} \sim \mathcal{D}} \left[ \log p_\theta(\boldsymbol{\tau}|y(\boldsymbol{\tau})) \right]. \tag{7}$$

Using a diffusion model to generate trajectories offers multiple advantages. First, the diffusion model leverages sequential relationships between consecutive transitions while generating trajectories, allowing it to minimize the degradation of dynamic plausibility. Second, the diffusion model captures long-term transition dynamics, which is beneficial for environments with sparse rewards.

**Diffusion Model Implementation**. Integrating sequential dependencies within the subtrajectory into the model architecture is essential for trajectory-level generation. Therefore, we choose a denoising network that combines Temporal-Unet [21] with MLP-mixer [33] to exploit both local and global sequential information. We adopted the Elucidated Diffusion Model [30], known for its powerful performance in recent work [13]. The implementation details are elaborated in Appendix A.2.

### 4.2 Stage B: Augment Trajectory-level Data

We propose a novel approach to augment trajectories, partial noising and denoising framework with amplified return guidance. Partial noising modifies the original trajectory using the forward process of the diffusion model, thus providing exploration opportunities. The exploration level is adjusted by the nosing ratio $\mu$. Following this, denoising with amplified return guidance refines the noised trajectory. During the denoising process, we guide the trajectory towards high-rewarding regions, promoting the exploitation of learned knowledge about the environment. We introduce the multiplier for conditioned return $\alpha$ to control the exploitation level. Figure 3 outlines the principle of data augmentation via partial noising and denoising framework.

**Partial noising with forward process**.

Let $\boldsymbol{\tau} = \boldsymbol{\tau}^0$ represent the original trajectory, and $k$ denotes the diffusion timestep. We introduce a noising ratio $\mu(0 < \mu = \frac{k}{K} \le 1)$ to determine the extent of erasing information of the trajectory for exploration. We add noise to the original trajectory $\boldsymbol{\tau}^0$, creating a noised trajectory denoted as $\boldsymbol{\tau}^{\mu \cdot K} \sim \mathcal{N}(\boldsymbol{\tau}; \boldsymbol{\tau}^0, \sigma(\mu \cdot K)^2 \mathbf{I})$. The parameter $\mu$ controls the level of exploration. Small $\mu$ results in minimal exploration, thereby preserving much of the original information of the trajectory. Large $\mu$ facilitates broader exploration, potentially leading to generating novel trajectories while losing original information significantly.

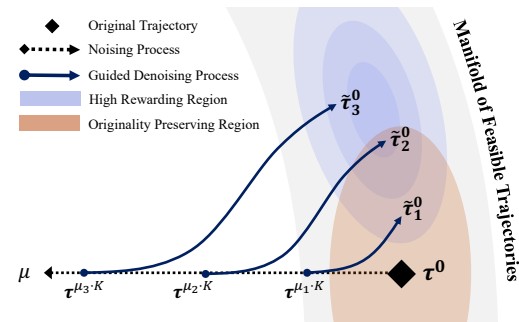

Figure 3: Mechanism of the *partial noising and denoising* framework. The extent of exploration increases with $\mu$ ($\mu_1 < \mu_2 < \mu_3$). During denoising, *amplified return guidance* shifts trajectories towards the high-rewarding region.

**Denoising with amplified return guidance**.

After noising the trajectory, we reconstruct the trajectory with classifier-free guidance to push the trajectory towards the high-rewarding region to enhance optimality. We introduce amplified return guidance, which sets the conditioning value as the multiplied return of the original trajectory. It can prevent adverse effects that occur when significantly larger return values are conditioned during generation. Further analysis on this phenomenon is elaborated in Appendix F.1.

The amplified return is formally defined as follows:

$$\hat{y}(\boldsymbol{\tau}^0, \alpha) = \alpha \cdot \sum_{i=t}^{T} \gamma^{i-t} r_i, \tag{8}$$

where $\alpha$ is the control parameter for the exploitation level. We set $\alpha > 1$ to make the conditioning value higher than the return of the original trajectory. $\alpha$ close to 1 induces mild conversion towards a high-rewarding trajectory region, while large $\alpha$ promotes significant drift, substantially exploiting the diffusion model. Our partial noising and denoising framework can be summarized as follows:

$$\text{Partial Noising} : (\boldsymbol{\tau}^0 \to \boldsymbol{\tau}^{\mu \cdot K} = \boldsymbol{\tau}^{\text{noised}}) \tag{9}$$

$$\text{Denoising} : (\boldsymbol{\tau}^{\mu \cdot K} \to \cdots \to \tilde{\boldsymbol{\tau}}^0 = \boldsymbol{\tau}^{\text{denoised}}) \tag{10}$$

### 4.3 Stage C: Offline RL Policy Training

The final stage of GTA is utilizing high-quality trajectories generated through previous stages for policy training. The trajectories generated by GTA can seamlessly integrate with existing offline RL methods. By introducing GTA, we can orthogonally integrate the expressiveness and controllability of diffusion models with existing offline RL algorithms.

### 4.4 Additional Technique: Reweighted Sampling

To concentrate on samples in high-rewarding regions, we additionally adopt a reweighting strategy [34, 35] during the sampling process. This approach prioritizes the sampling of subtrajectories with higher returns. Consequently, augmented trajectories are more densely distributed in high-rewarding regions. The complete specification of our reweighting strategy, including implementation details and parameter configurations, is provided in Appendix A.3.

## 5 Experiments

In this section, we present extensive experiments conducted across commonly studied benchmarks [15, 16] to evaluate the augmentation capability of GTA. Through experiments, we aim to address the following questions: **1)** How much performance gain does GTA exhibit across various types of tasks? **2)** What impacts do the design elements of GTA have on performance? **3)** Are trajectories augmented via GTA high quality? **4)** What is the preferable $\mu$ and $\alpha$ setting for the new tasks?

### 5.1 Experimental Setup

**Datasets and environments**. We demonstrate the versatility of GTA on the D4RL benchmark [15], from the standard continuous control tasks to the challenging tasks with unique difficulties. We validate that GTA enhances offline RL under the sparse rewards environments and overcoming high-dimensional, complex robotics tasks. Additionally, we demonstrate that GTA is effective with human-demonstrated, small-scale datasets, showing its adaptability to realistic settings. Finally, we extend GTA to pixel-based environments, specifically within VD4RL benchmark [16].

**Data augmentation baselines**. We compare GTA with existing augmentation methods. For traditional augmentation, we choose S4RL [12], which introduces Gaussian noise into states. In the domain of generative augmentation, we choose SynthER [13], which employs an unconditional diffusion model for transition-level generation.

**Offline RL algorithms**. To demonstrate the general efficacy of GTA on proprioceptive observations, we select four widely used offline RL algorithms: TD3BC [7], CQL [8], IQL [36], and MCQ [4]. For particularly challenging tasks, such as Maze2d, Antmaze, Adroit, and FrankaKitchen, we employ IQL, which provides all hyperparameter configurations across all tasks and consistently demonstrates stable performance. For pixel-based observations, we utilize DrQ+BC as the baseline algorithm, as established in the VD4RL benchmark by Lu et al. [16].

**Data quality metrics**. We propose data quality metrics to analyze whether GTA provides high-quality data. Following prior works [13, 37], we conduct a thorough analysis with three metrics: oracle reward, novelty, and dynamic MSE. Oracle reward, computed with the true reward of generated data, represents optimality. Novelty measures the ability of the augmentation method to discover novel states and actions not existing in the offline data. Finally, dynamic MSE evaluates how well the generated trajectories adhere to the dynamics of the environment. Formally, dynamic MSE and novelty are defined as follows:

$$\text{Dynamic MSE}(\mathcal{D}_{\text{A}}) = \frac{1}{|\mathcal{D}_{\text{A}}|} \sum_{(s,a,r,s') \in \mathcal{D}_{\text{A}}} (f^*(s,a) - s')^2 \tag{11}$$

$$\text{Novelty}(\mathcal{D}_{\text{A}}, \mathcal{D}) = \frac{1}{|\mathcal{D}_{\text{A}}|} \sum_{(s,a,r,s') \in \mathcal{D}_{\text{A}}} \min_{(\bar{s},\bar{a},\bar{r},\bar{s}') \in \mathcal{D}} ((s,a) - (\bar{s},\bar{a}))^2 \tag{12}$$

where $\mathcal{D}_{\text{A}}$ represent the augmented dataset and $\mathcal{D}$ is original offline dataset. $(s, a, r, s')$ denotes a single transition, and $f^*$ denotes the true dynamic model of the environment.

Table 1: Normalized average scores on Gym locomotion and maze tasks, with the highest scores highlighted in **bold**. Each cell displays the mean and standard deviation across 8 seeds.

| Algo. | Aug. | Halfcheetah | | | Hopper | | | Walker2d | | | Average |
|---|---|---|---|---|---|---|---|---|---|---|---|
| | | medium | medium-replay | medium-expert | medium | medium-replay | medium-expert | medium | medium-replay | medium-expert | |
| TD3BC | None | 48.42 ± 0.62 | 44.64 ± 0.71 | 89.48 ± 5.50 | 61.04 ± 3.18 | 65.69 ± 24.41 | 104.08 ± 5.81 | 84.58 ± 1.92 | 84.11 ± 4.12 | 110.23 ± 0.37 | 76.92 ± 2.66 |
| | S4RL | 48.74 ± 0.31 | 44.53 ± 0.30 | 90.78 ± 4.65 | 59.34 ± 3.50 | 67.39 ± 23.81 | 106.10 ± 7.24 | 84.63 ± 2.44 | 83.42 ± 4.70 | 110.21 ± 0.35 | 77.24 ± 3.00 |
| | SynthER | 49.16 ± 0.39 | 45.57 ± 0.34 | 85.47 ± 11.35 | 63.70 ± 3.69 | 78.81 ± 15.80 | 98.99 ± 11.27 | 85.43 ± 1.14 | 90.67 ± 1.56 | 109.95 ± 0.32 | 78.64 ± 2.38 |
| | GTA | **57.84 ± 0.51** | **50.04 ± 0.84** | **93.13 ± 3.07** | **69.57 ± 4.05** | **89.31 ± 16.84** | **110.40 ± 4.04** | **86.69 ± 0.89** | **93.82 ± 1.74** | **110.86 ± 0.34** | **84.63 ± 2.20** |
| CQL | None | 46.98 ± 0.20 | 44.70 ± 0.51 | **95.90 ± 0.52** | 61.13 ± 3.20 | 82.33 ± 16.37 | 104.38 ± 7.30 | **82.26 ± 1.18** | 79.74 ± 5.19 | 109.50 ± 0.43 | 78.55 ± 1.88 |
| | S4RL | 47.00 ± 0.23 | 44.62 ± 0.42 | 95.89 ± 0.45 | 62.72 ± 3.46 | 78.82 ± 9.89 | 108.87 ± 2.69 | 81.46 ± 2.28 | 80.82 ± 8.35 | 109.65 ± 0.22 | 78.87 ± 1.62 |
| | SynthER | 47.21 ± 0.14 | 46.03 ± 0.40 | 95.29 ± 1.90 | 64.65 ± 4.78 | 92.06 ± 13.40 | 107.66 ± 6.68 | 81.91 ± 0.89 | 86.62 ± 3.03 | 109.36 ± 0.36 | 81.20 ± 1.46 |
| | GTA | 54.14 ± 0.31 | **51.36 ± 0.27** | 94.93 ± 3.71 | **74.80 ± 7.42** | **98.88 ± 3.51** | **110.90 ± 3.44** | 80.40 ± 4.98 | **91.57 ± 5.15** | **110.44 ± 0.28** | **85.27 ± 1.02** |
| IQL | None | 48.65 ± 0.19 | 43.35 ± 0.50 | 94.57 ± 1.88 | 66.35 ± 7.09 | 95.76 ± 4.01 | 91.69 ± 25.97 | 84.34 ± 3.31 | 69.60 ± 10.80 | 112.37 ± 0.60 | 78.52 ± 3.52 |
| | S4RL | 48.58 ± 0.29 | 43.57 ± 0.65 | 94.22 ± 1.59 | 65.06 ± 5.94 | 86.72 ± 22.01 | 99.82 ± 8.09 | **84.58 ± 4.26** | 70.33 ± 7.99 | 112.29 ± 0.79 | 78.35 ± 2.25 |
| | SynthER | 49.76 ± 0.27 | **46.91 ± 0.28** | 91.90 ± 3.75 | 69.21 ± 5.85 | **102.97 ± 1.65** | 94.08 ± 23.94 | 80.15 ± 16.47 | 90.63 ± 4.66 | 112.12 ± 0.53 | 81.97 ± 2.21 |
| | GTA | **54.82 ± 0.35** | 46.89 ± 3.00 | **95.30 ± 0.55** | **77.46 ± 3.42** | 102.11 ± 1.51 | **107.78 ± 4.66** | 84.40 ± 2.32 | **93.37 ± 6.35** | **112.87 ± 0.66** | **86.11 ± 0.94** |
| MCQ | None | 60.98 ± 0.72 | 54.09 ± 0.96 | **80.51 ± 9.45** | **73.97 ± 9.57** | 101.86 ± 0.84 | 92.85 ± 17.35 | 89.70 ± 2.66 | 92.19 ± 1.89 | **114.55 ± 2.10** | 84.52 ± 1.84 |
| | S4RL | 60.93 ± 0.88 | 53.32 ± 0.99 | 79.02 ± 16.25 | 73.95 ± 10.68 | 100.62 ± 1.84 | 83.91 ± 30.39 | 90.91 ± 0.82 | 89.90 ± 5.44 | 102.62 ± 33.46 | 81.69 ± 3.99 |
| | SynthER | 63.57 ± 0.69 | **58.34 ± 1.03** | 45.64 ± 7.56 | 65.61 ± 21.89 | **103.54 ± 0.83** | **106.02 ± 7.71** | **92.31 ± 1.04** | 98.02 ± 3.51 | 112.22 ± 1.08 | 82.81 ± 3.21 |
| | GTA | **63.76 ± 0.40** | 54.17 ± 2.23 | 80.00 ± 9.03 | 66.75 ± 11.64 | 102.19 ± 1.09 | 99.27 ± 13.48 | 91.22 ± 3.25 | **104.24 ± 1.90** | 111.63 ± 1.42 | **85.91 ± 1.05** |

| Algo. | Aug. | Maze2d | | | AntMaze | | | | | | Average |
|---|---|---|---|---|---|---|---|---|---|---|---|
| | | umaze | medium | large | umaze | medium-play | large-play | umaze-diverse | medium-diverse | large-diverse | |
| IQL | None | 37.41 ± 2.83 | 32.80 ± 1.49 | 58.99 ± 9.16 | 58.75 ± 8.90 | 78.13 ± 3.44 | 40.63 ± 8.75 | 50.38 ± 17.39 | 65.50 ± 9.46 | 45.75 ± 6.34 | 52.04 ± 3.89 |
| | S4RL | 37.69 ± 3.36 | 34.82 ± 3.16 | 62.93 ± 3.47 | 55.00 ± 10.47 | 80.88 ± 5.17 | 42.88 ± 8.71 | 51.63 ± 11.67 | 74.00 ± 9.72 | 46.13 ± 8.34 | 53.99 ± 2.82 |
| | SynthER | 39.00 ± 2.26 | 34.27 ± 2.51 | 61.74 ± 4.51 | 17.13 ± 6.45 | 41.00 ± 20.58 | 37.50 ± 6.48 | 23.94 ± 11.83 | 40.88 ± 14.15 | 37.50 ± 8.37 | 36.99 ± 2.98 |
| | GTA | **41.68 ± 1.41** | **37.78 ± 1.66** | **76.56 ± 4.70** | **66.50 ± 6.91** | **81.88 ± 4.19** | **44.38 ± 4.66** | **57.88 ± 9.51** | **78.13 ± 7.85** | **47.75 ± 6.69** | **62.75 ± 1.03** |

Table 2: Normalized average scores on complex robotics tasks , with the highest scores highlighted in **bold**. Each cell displays the mean and standard deviation across 8 seeds.

| Algo. | Aug. | Adroit | | Average | FrankaKitchen | | | Average |
|---|---|---|---|---|---|---|---|---|
| | | pen-human | door-human | | partial | mixed | complete | |
| IQL | None | 69.52 ± 5.48 | 3.34 ± 1.16 | 36.43 ± 2.97 | 38.06 ± 3.15 | 54.88 ± 2.68 | 57.75 ± 4.33 | 50.23 ± 1.45 |
| | S4RL | 72.52 ± 5.79 | 3.22 ± 0.80 | 37.87 ± 3.22 | 36.78 ± 2.30 | 54.25 ± 3.26 | 55.06 ± 3.52 | 48.7 ± 1.40 |
| | SynthER | 72.13 ± 4.48 | 3.77 ± 0.72 | 37.95 ± 2.13 | 37.38 ± 1.55 | 56.13 ± 0.61 | **59.04 ± 6.98** | 50.85 ± 6.30 |
| | GTA | **76.11 ± 9.54** | **9.35 ± 1.48** | **42.73 ± 4.26** | **45.91 ± 6.41** | **56.22 ± 1.88** | 57.78 ± 3.58 | **53.3 ± 3.23** |

## 5.2 Benchmark Evaluations

We conducted experiments to evaluate whether GTA can provide a performance boost when applied alongside existing offline RL algorithms in various tasks with its own unique challenges.

**Gym locomotion**. We experimentally demonstrate on Table 1 that offline RL algorithms with GTA outperform all other baselines across all algorithms on the average score with a statistically significant margin. This result highlights the versatility of GTA with various offline RL algorithms. We provide $p$-value for our experiments to validate the statistical significance of the results in Appendix F.2.

**Maze tasks**. To assess GTA in sparse reward tasks, we test GTA on Maze2d and AntMaze tasks. Table 1 reveals that GTA notably enhances the performance of IQL policy on the sparse reward tasks while SynthER often degrades the performance. This result suggests two insights. First, the trajectory-level generation of GTA helps capture long-term dynamics, effectively leveraging that information while augmenting sparsely rewarded trajectories. Second, augmenting the dataset with high-rewarding trajectories strengthens the goal-reaching ability by enriching the demonstrations with more successful trajectories.

**Complex robotics tasks**. We evaluate the effectiveness of the GTA on realistic, high-dimensional, challenging control tasks. Adroit-human datasets comprise 25 human-generated trajectories. The FrankaKitchen dataset, which involves multitasking behavior trajectories, requires a generalization ability to stitch trajectories. According to the results in Table 2, GTA effectively boosts the performance in high-dimensional, complex robotics tasks. The results on the Adroit-human dataset demonstrate that GTA effectively augments small-scale human demonstration data through the expressiveness of the diffusion model. Additionally, the performance improvements in kitchen tasks indicate that GTA aids offline RL in trajectory stitching [21].

**Pixel-based Observations**. We extend our approach to pixel-based observations by following the experimental setup detailed in [13]. Initially, we pretrain the policy and extract its visual encoder. Subsequently, we project offline pixel-based observations into the embedding space, followed by augmenting these embedded states with GTA. More detailed experiment setups are elaborated in Appendix C.4. The results presented in Table 3 show performance improvements on DrQ+BC across the environments and dataset qualities. It indicates that GTA can be further extended to pixel-based observations beyond proprioceptive observations.

Table 3: Normalized average scores on pixel-based observation tasks, with the highest scores highlighted in **bold**. Each cell displays the mean and standard deviation across 6 seeds.

| Algo. | Aug. | Cheetah-run | | | | Walker-walk | | | | Average |
|---|---|---|---|---|---|---|---|---|---|---|
| | | medium | medium-replay | medium-expert | expert | medium | medium-replay | medium-expert | expert | |
| DrQ+BC | None | **53.3 ± 3.0** | **44.8 ± 3.6** | 50.6 ± 8.2 | 34.5 ± 8.3 | **40.1 ±3.1** | 13.4 ±4.0 | 43.3 ±4.1 | 63.3 ±5.1 | 42.7 ±4.1 |
| | SynthER | 50.4 ± 3.1 | 25.6 ± 2.9 | 36.8 ± 6.0 | 11.9 ± 2.4 | 29.1 ±4.4 | 7.6 ±2.5 | 30.2 ±2.7 | 28.7 ±3.8 | 27.5 ±3.3 |
| | GTA | **53.3 ± 1.9** | 38.1 ± 2.5 | **61.0 ± 8.2** | **46.8 ± 8.2** | 38.0 ±5.0 | **13.6 ±1.5** | **45.3 ±5.7** | **67.4 ±10.5** | **45.4 ±5.7** |

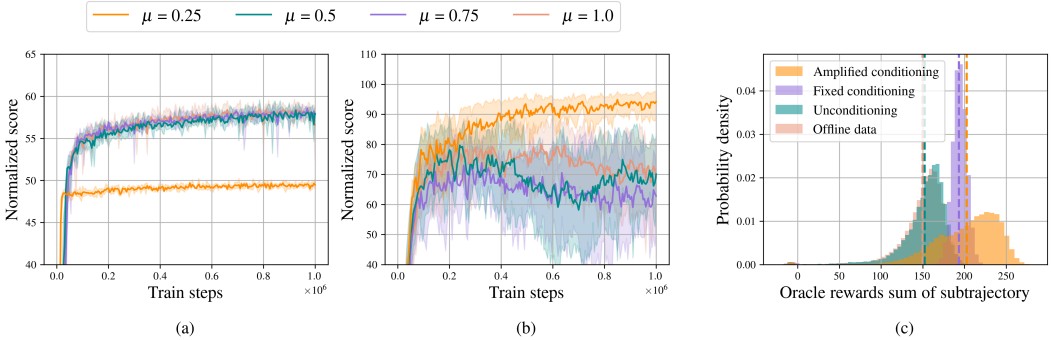

Figure 4: (a), (b) D4RL normalized score across different noise levels over the course of training TD3BC on halfcheetah-medium-v2 and halfcheetah-medium-expert-v2. (c) Comparison oracle reward sum of subtrajectory between conditioning strategy on halfcheetah-medium-v2.

## 5.3 Ablation Studies

We carry out thorough ablation studies to assess the effectiveness of each component within GTA. These studies are evaluated based on the performance scores of D4RL tasks and the aforementioned data quality metrics.

**What are the benefits of trajectory-level generation?** To investigate the impact of utilizing sequential relationships through trajectory-level generation, we examine performance differences as the trajectory length varied. We find that when $H = 1$, the dynamic MSE becomes five times or more higher compared to trajectory-level generation with $H$ longer than 16. Additionally, when training TD3BC with augmented data, the normalized score increased by $21\%$ with trajectory generation compared to transition generation, where $H = 1$. This result highlights sequential relationship between conseuive transitions in trajectory does help generation, minimizing degradation of dynamic plausibility. Detailed experiment results are in Appendix E.1

**Is preserving original information via partial noising essential?** We conduct experiments by varying the noising ratio on datasets of different optimality to explore the effect of the exploration on performance. In Figure 4a halfcheetah-medium, higher noise levels enhance performance, while in Figure 4b halfcheetah-medium-expert, optimal performance occurs at a noise level of $\mu = 0.25$, with declines at higher levels. We speculate that if the original data already contains high-rewarding trajectories, even minimal modifications can be highly effective, and excessive exploration might lead to unexpected outcomes. Therefore, preserving information of original trajectory by partial noising is crucial in achieving effective data augmentation while avoiding potential negative impacts from excessive exploration. Additionally, the terminal signal, which may be very sparse, could be lost significantly if the original information is not preserved. We further conduct experiments about the impact of partial noising on preserving terminal state information in Appendix F.3.

**Does amplified return guidance improve the optimality?** Figure 4c illustrates the superiority of the amplified return guidance on augmentation with respect to unconditioning [13], and fixed conditioning [22]. Augmented trajectories without any condition tend to replicate the original reward distribution, thereby failing to enhance data optimality. Fixed conditioning concentrates rewards near the maximum return offline dataset. Notably, amplified return conditioning demonstrates a superior ability to cover a broader range of rewards, especially on unseen high rewards, while leading on the average reward and D4RL score. Further analysis of the impact of the conditioning method on data quality can be found in Appendix F.1.

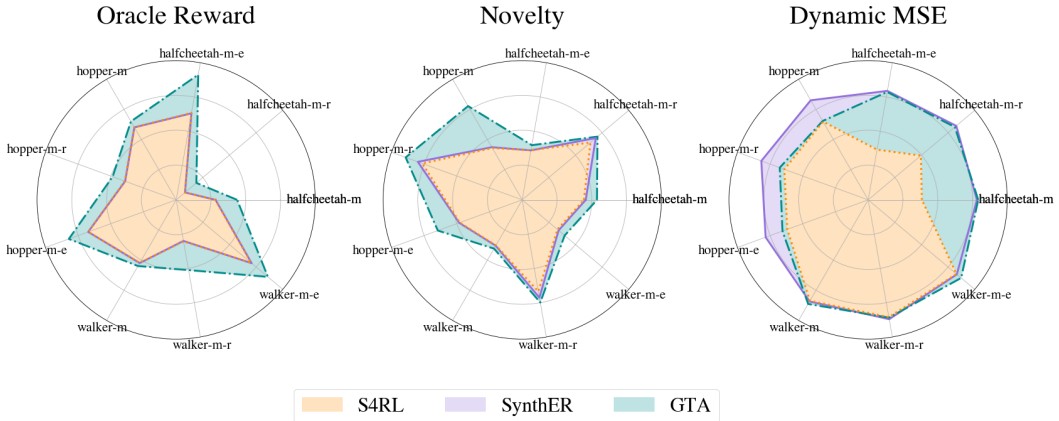

Figure 5: Data quality analysis of S4RL, SynthER, and GTA. GTA augmented data exhibits superior optimality and novelty across gym locomotion datasets while maintaining dynamic plausibility.

## 5.4 Data Quality Analysis

We introduce quality metrics specifically suitable for offline data analysis in Section 5.1. We compare GTA with other augmentation methods regarding not only how much it expands the coverage of offline data but also its optimality and dynamic MSE. Figure 5 presents the relative scale of each metric across baseline augmentation methods on gym locomotion tasks. Data augmented by GTA shows significantly higher oracle reward and novelty while maintaining a comparably low dynamic MSE. These outcomes indicate that the conditional diffusion model of GTA expands data coverage under dynamic constraints and discovers novel states and actions with higher rewards, effectively integrating environmental knowledge. We add further analysis in Appendix F.4.

## 5.5 Guideline for Selecting $\mu$ and $\alpha$

Table 4: D4RL normalized score on locomotion environments with fixed $\alpha$ and $\mu$. The experiments are conducted with TD3BC.

| Algo. | Halfcheetah | | | Hopper | | | Walker2d | | | Average |
|---|---|---|---|---|---|---|---|---|---|---|
| | medium | medium-replay | medium-expert | medium | medium-replay | medium-expert | medium | medium-replay | medium-expert | |
| None | 48.42 ± 0.62 | 44.64 ± 0.71 | 89.48 ± 5.50 | 61.04 ± 3.18 | 65.69 ± 24.41 | 104.08 ± 5.81 | 84.58 ± 1.92 | 84.11 ± 4.12 | 110.23 ± 0.37 | 76.92 ± 2.66 |
| SynthER | 49.16 ± 0.39 | 45.57 ± 0.34 | 85.47 ± 11.35 | 63.70 ± 3.69 | **78.81 ± 15.80** | 98.99 ± 11.27 | **85.43 ± 1.14** | **90.67 ± 1.56** | 109.95 ± 0.32 | 78.64 ± 2.38 |
| **GTA** ($\mu = 0.25, \alpha = 1.1$) | **49.22 ± 0.52** | **46.48 ± 0.39** | **94.98 ± 1.66** | **66.16 ± 4.89** | 72.86 ± 28.42 | **107.09 ± 2.69** | 85.42 ± 1.30 | 86.02 ± 8.98 | **110.67 ± 0.89** | **79.88 ± 3.35** |

Table 5: D4RL normalized score on medium and medium-replay quality locomotion environments with fixed $\alpha$ and $\mu$. The experiments are conducted with TD3BC.

| Algo. | Medium | | | Medium-replay Environment | | | Average |
|---|---|---|---|---|---|---|---|
| | Halfcheetah | Hopper | Walker2d | Halfcheetah | Hopper | Walker2d | |
| None | 48.42 ± 0.62 | 61.04 ± 3.18 | 84.58 ± 1.92 | 44.64 ± 0.71 | 65.69 ± 24.41 | 84.11 ± 4.12 | 64.75 ± 4.15 |
| SynthER | 49.16 ± 0.39 | 63.70 ± 3.69 | 85.43 ± 1.14 | 45.57 ± 0.34 | 78.81 ± 15.80 | 90.67 ± 1.56 | 68.89 ± 2.62 |
| **GTA** ($\mu = 0.5, \alpha = 1.3$) | **57.92 ± 0.48** | **68.46 ± 1.32** | **88.38 ± 2.70** | 48.23 ± 5.42 | 77.17 ± 22.17 | **91.12 ± 6.45** | 72.17 ± 11.23 |
| **GTA** ($\mu = 0.75, \alpha = 1.3$) | 57.85 ± 0.27 | 61.58 ± 5.00 | 87.14 ± 1.73 | **48.99 ± 2.16** | **97.26 ± 3.38** | 90.89 ± 3.29 | **79.05 ± 2.90** |

In this section, we propose a guideline for selecting hyperparameters based on dataset quality and environment characteristics. First, when the known information about the dataset is limited and minimizing exploration risk is essential, a low exploration and exploitation approach can enhance performance without losing stability. As illustrated in Table 4, GTA with low exploration and exploitation ($\mu = 0.25, \alpha = 1.1$) surpasses baseline methods in gym locomotion tasks.

However, if we know that the offline dataset is suboptimal and there exists considerable room for improvement, a higher exploration and exploitation setting can yield superior performance by uncovering a broader range of high-quality behaviors. Table 5 shows that GTA with ($\mu = 0.5, \alpha = 1.3$) and ($\mu = 0.75, \alpha = 1.3$) significantly boosts performance on suboptimal datasets. These findings indicate that while conservative parameters with low $\mu$ and $\alpha$ offer stable gains, we can elevate values of $\mu$ and $\alpha$ by leveraging prior knowledge about dataset quality to achieve superior outcomes. Notably, GTA consistently outperforms baselines across all configurations without dataset-specific hyperparameter tuning.

### 5.6 Futher Experimental Results

We conduct extensive additional experiments to explore the further potential of the GTA.

**GTA under realistic settings**. We examined the augmentation ability of GTA for two realistic settings: one with a mixed dataset consisting of few expert datasets and the majority of low-performing trajectories in Appendix E.2, and the other with a small amount of offline dataset in the Appendix E.3. In both cases, We observed that GTA significantly enhances the performance, improving sample efficiency of the offline RL.

**Larger batch size and training epochs**. Since GTA offers more training data, it is worth exploring whether increasing batch size or training epochs can lead to additional performance gain. The results in Appendix G.1 demonstrate that we can enhance the performance by increasing batch size and training epochs while baselines do not benefit from the batch size and training epochs increment.

**Extending to sequence modeling approach**. We proved that GTA can be applied to sequence modeling approaches such as Decision Transformer [38] as GTA augments data at the trajectory level. We confirmed that GTA improves the performance of the sequence modeling approach, especially in sparse reward tasks. The detailed experiment setup and results are in Appendix E.4.

**Sensitivity tests**. We conduct experiments on the sensitivity of parameters $\alpha$ in Appendix G.2 and $\mu$ in Appendix G.3. The sensitivity of GTA with the amount of augmented dataset is in Appendix G.4. Our results indicate that GTA maintains similar performance beyond one million augmented transitions, demonstrating robustness to the number of transition samples.

**Additional ablations on design choices**. We explore the impact of reweighted sampling introduced on Section 4.4 in Appendix E.5. For exploring alternative design choices, we adopt conditioning-by-inpainting as an alternative conditioning approach instead of classifier-free guidance in Appendix E.6. We also test the model-based approach by replacing the reward of the generated trajectory with the prediction of the reward proxy model in Appendix E.7.

## 6 Conclusion

We propose **GTA**: Generative Trajectory Augmentation, a novel generative data augmentation approach for enriching offline data with high-rewarding and dynamically plausible trajectories. GTA combines data augmentation with the diffusion sampling process by partially adding noise to original trajectories and subsequently denoising them under amplified return guidance. Our extensive experiments on 31 datasets across 9 environments exhibit considerable performance improvements in four distinct offline RL algorithms, demonstrating the versatility and effectiveness of the GTA. We show that GTA successfully creates high-quality datasets from the sub-optimal offline dataset, which leads to learning effective decision-making policy.

**Limitations and future works**. In the GTA framework, we do not train additional transition model and reward model for simple augmentation framework. Therefore, in tasks where dynamic violations have a critical impact, we may not expect a significant performance boost. However, as shown in our experiments, dynamic violations are generally minimal. Additionally, while the main focus of GTA is on the offline setting, future work could explore off-to-online and online settings.

## Acknowledgement

We thank the anonymous reviewers for their insightful comments and suggestions, which significantly improve our manuscript. This work was supported by the National Research Foundation of Korea(NRF) grant funded by the Korea government(MSIT) (No. RS-2024-00410082).

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

# Appendix

We provide further details of our paper in the appendix. Our code implementation can be found in `https://github.com/Jaewoopudding/GTA`

## A  Methodology Details

### A.1  Subtrajectory Generation Details

In this section, we provide further details for our trajectory-level generation. As generating a whole episode that includes more than hundreds of timesteps at once is infeasible, we generate a subtrajectory with a horizon $H$, following prior works [21, 22]. Each subtrajectory consists of a sequence of state, action, and reward.

Note that we need full environment transitions, i.e., states, actions, rewards, next states, and terminals (if MDP is finite) for training conventional Q-learning-based Offline RL algorithms. To achieve this, we actually model the subtrajectory of length $H + 1$ and use a generated state $s_{t+H}$ for the next state of $H$th transition.

$$\boldsymbol{\tau} = \begin{bmatrix} s_t & s_{t+1} & \cdots & s_{t+H-1} & s_{t+H} \\ a_t & a_{t+1} & \cdots & a_{t+H-1} & \emptyset \\ r_t & r_{t+1} & \cdots & r_{t+H-1} & \emptyset \end{bmatrix}$$
$$\longrightarrow \mathcal{D} = \{(s_t, a_t, s_{t+1}, r_t), \cdots, (s_{t+H-1}, a_{t+H-1}, s_{t+H}, r_{t+H-1})\}$$

In environments like Walker2d and Hopper, which have terminal transitions, each trajectory may have a different episode length. It is necessary to incorporate terminal information when setting the conditioning value for guiding generative augmentation. When it comes to directly generating terminal value, modeling the terminal variable is hard due to its binary nature and sparseness, as pointed out in [13]. The situation is worse in trajectory-level generation as it should be forbidden that the terminal variable is set to 1 in intermediate transitions. To address this, we follow the implementation in [22] by adjusting the return calculation and generation process: we subtract 100 from the reward of terminal transitions. This modification allows for the encoding of terminal information without the need for binary variable terminal generation.

### A.2  Model Architecture Details

We represent the denoising network model with a temporal U-Net [22] with 6 repeated residual blocks. We choose MLP-Mixer [33] architecture for modeling each block instead of temporal convolution layers. In MLP-Mixer, we first concatenate each component (states, actions, and rewards) and project it into the MLP layer dimension-wise, then apply MLP across temporal locations and repeat the process. By doing this, we can efficiently capture both temporal and component-wise relationships.

### A.3  Reweighted Sampling

We adopt reweighted sampling, which is suggested in [34, 35]. First, we segment the condition $y$, which is the return value of subtrajectory, into $N_B$ equal intervals. Each interval is weighted according to its value and the number of points in the interval, with the specific weight $v_j$ calculations applied as follows:

$$v_j = \frac{|B_j|}{|B_j| + u} \exp\left(\frac{-|\hat{y} - y_{b_j}|}{q}\right), \tag{13}$$

where $\hat{y}$ represents the maximum return value in the offline dataset $\mathcal{D}$, $|B_j|$ denotes the count of data points in the $j^{th}$ bin, and $y_{b_j}$ refers to the midpoint of the interval associated with bin $B_j$. In the equation, $u$ and $q$ serve as smoothing parameters. The parameter $q$ determines the weighting between high-score bins and low-score bins. Lowering $q$ results in reduced weights for the low-score bins, and conversely, increasing $q$ assigns higher weights to them. The detail setting of the $u$ and $q$ is provided in Appendix C.3.

# B  Hyperparameters

## B.1  Diffusion Models

**Size of the diffusion model and training details**. We use the same size of network and hyperparameters for modeling conditional diffusion model across all experiments for ease of implementation. The hyperparameters we used for modeling and training are listed in Table 6. The number of trainable parameters of our diffusion model is about 29.6M, which is comparable with the size of the diffusion model used in SynthER $\approx$ 25.6M. When using the default network model, training takes approximately 30 hours for 1M training steps with batch size 256 on a 4 RTX3090 NVIDIA GPU.

Table 6: Hyperparameter setting for backbone of diffusion model for GTA

|  | Parameters | Values |
|---|---|---|
|  | Batch size | 256 |
|  | Optimizer | Adam |
|  | Learning Rate | $3 \times 10^{-4}$ |
| Training | Learning Rate Schedule | Cosine Annealing Warmup |
|  | Training Steps | 1e6 |
|  | Conditional dropout ($\lambda$) | 0.25 |

**Trajectory sampling from the diffusion model**. For the sampling process, we use the Stochastic Differential Equation (SDE) sampler suggested in EDM [30] with the default hyperparameter setting used in SynthER [13]. As we employ classifier-free guidance, an additional hyperparameter guidance scale $w$ has been introduced. We use the same guidance scale $w = 2.0$ for all experiments. Detailed hyperparameters are listed in Table 7.

Table 7: Hyperparameter setting of diffusion model of GTA for trajectory sampling

|  | Parameters | Values |
|---|---|---|
|  | Number of Diffusion Steps | 128 |
|  | $\sigma_{\min}$ | 0.002 |
|  | $\sigma_{\max}$ | 80 |
| EDM | $S_{\mathrm{churn}}$ | 80 |
|  | $S_{\min}$ | 0.05 |
|  | $S_{\max}$ | 50 |
|  | $S_{\mathrm{noise}}$ | 1.003 |
| Conditioning | Guidance scale ($w$) | 2.0 |

To sample 5M transitions, it takes approximately 3.5 hours with 128 diffusion timesteps on a single RTX 3090 NVIDIA GPU. Note that our sampling time can be reduced as we adopt the partial noising and denoising techniques. For example, with $\mu = 0.5$, we only require 64 denoising steps and can accelerate the sampling process.

## B.2 Detailed parameter settings in GTA

We provide hyperparameter search spaces for partial noising level $\mu$, multiplier $\alpha$, horizon length $H$, and reweighting parameter used for GTA in Table 8.

Table 8: Hyperparameter setting for partial noising level $\mu$ and guidance multiplier $\alpha$

| Environment | $\alpha$ | $\mu$ | H | Reweighting |
|---|---|---|---|---|
| Locomotion | {1.1, 1.3, 1.4} | {0.5, 0.25} | {32, 128} | {$\varnothing$, $N_B$ = 20, 50, 100} |
| Maze2d | {1.25, 1.3} | {0.5} | {32} | {$c$ = 10} |
| AntMaze | {1.1, 1.3, 1.5} | {0.05, 0.1, 0.25} | {32} | {$\varnothing$, $c$ = 10, 30} |
| Adroit | {1.1 ,1.3} | {0.1, 0.25} | {32} | {$\varnothing$} |
| FrankaKitchen | {1.1, 1.3, 1.5} | {0.1, 0.25, 0.5} | {32} | {$\varnothing$, $c$ = 10} |

Note that in sparse reward settings where the reward is a binary variable, the reweighted strategy tends to predominantly sample subtrajectories near the goal location. It might not be helpful since it makes the distribution of generative data cover a narrow region. To address this, we sample episodes instead of subtrajectories with reweighting in sparse reward settings. We assign the weight $c$ times the bigger value to the successful episodes. Once the episode is sampled via weighted sampling, subtrajectories for augmentation is sampled from the uniform distribution. For dense reward environments like Gym locomotion, we put details of reweighting parameters in Table 9

Table 9: Hyperparameter setting for reweighted sampling

| Environment | Reweighting | $N_B$ | $u$ | $q$ |
|---|---|---|---|---|
| Halfcheetah-medium-v2 | X | - | - | - |
| Halfcheetah-medium-replay-v2 | O | 50 | 0.001 | 5.0 |
| Halfcheetah-medium-expert-v2 | O | 20 | 0.001 | 5.0 |
| Hopper-medium-v2 | O | 50 | 0.001 | 5.0 |
| Hopper-medium-replay-v2 | O | 50 | 0.001 | 5.0 |
| Hopper-medium-expert-v2 | O | 50 | 0.001 | 5.0 |
| Walker2d-medium-v2 | O | 100 | 0.001 | 5.0 |
| Walker2d-medium-replay-v2 | O | 50 | 0.001 | 5.0 |
| Walker2d-medium-expert-v2 | O | 50 | 0.001 | 5.0 |

# C  Experiment Details

## C.1  Data Augmentation Baselines

In this subsection, we describe the implementation details of the data augmentation baselines. Note that we try to follow the official implementation provided by authors, including hyperparameters, and re-implement only when public code is unavailable.

- **S4RL [12]**: S4RL adds zero-mean Gaussian noise to input states as augmented data, i.e, $\tilde{s}_t \leftarrow s_t + \epsilon, \ \epsilon \sim \mathcal{N}(0, \sigma I)$. We choose $\sigma = 0.0003$, following the original paper.

- **SynthER [13]**: SynthER trains an unconditional diffusion model that generates transition-level data (state, action, reward, next state, and terminals). We build SynthER based on code provided by authors[2]. For a fair comparison, we re-implement the setting for the offline RL with a larger diffusion network for all environments and sample the same number of transitions. The diffusion model of SynthER contains 25.6M trainable parameters. We only use generated data from SynthER to train offline RL methods for evaluation, following the original paper. Their code is opened at `https://github.com/conglu1997/SynthER`

## C.2  Offline RL Algorithms

We employed Clean Offline Reinforcement Learning (CORL) library [39] and OfflineRL-Kit library [40] to adapt implementation code and hyperparameters of the Offline RL algorithms. We import CQL [8], IQL [36], TD3BC [7] and DT [38] from the CORL library, and MCQ [4] from the OfflineRL-Kit. The codebase of the CORL can be found at `https://github.com/tinkoff-ai/CORL`, and OfflineRL-Kit from `https://github.com/yihaosun1124/OfflineRL-Kit`.

## C.3  Dataset Version and Evaluation Procedure

We employ v2 dataset for the Gym locomotion and AntMaze tasks, v1 for the Maze2d tasks, and v0 for the Adroit and FrankaKitchen tasks. We follow the standard evaluation procedure of D4RL benchmarks. For all the tasks we examined, we train TD3BC, IQL, CQL, MCQ for 1M training steps and report the score over the final evaluations. Especially, in case of the DT in Appendix E.4, we trained it for 100k steps. We reported the final evaluation scores of the experiments, using 10 evaluation steps for Gym locomotion tasks and 100 steps for Maze2d, AntMaze, Adroit, and Franka Kitchen tasks.

We normalize the score where 0 corresponds to a random policy, and 100 corresponds to an expert policy as proposed at D4RL benchmark [15]. All experiments of the main sections are conducted over 8 different random seeds. We set the batch size as 1024 across all Q-learning based offline RL algorithms. The justification for batch size configuration is covered at Appendix G.1

## C.4  Pixel-based Environments

We conducted experiments on pixel-based environments using the VD4RL benchmark [16]. We follow the experiment protocol of the pixel-based experiment of Synther [13], firstly pretrain the CNN encoder with image observations for 1M steps and then fine-tune policy and Q-network with the augmented latent observations for 300k steps. We reported the mean and standard deviation of the last evaluation score. We calculate the standard deviation of the average score by averaging the standard deviation of each dataset, not identical with the standard deviation calculate protocol of Table 1 and Table 2, which utilize the standard deviation of mean performance of each seed. We used the DrQ+BC codebase and the offline dataset on the `https://github.com/conglu1997/v-d4rl` and re-implement the fine-tuning.

We used $\alpha = 2.0$ and $\mu = 0.1$ for the medium, medium-expert, and expert dataset and $\alpha = 1.3$ and $\mu = 0.1$ for the medium-replay dataset. For the reweighted sampling, we choose $N_B = 50$, $u = 0.001$, and $q = 5.0$ across all datasets.

---

[2]https://github.com/conglu1997/SynthER

## D   Data Quality Metrics

In this section, we present details about the quality metrics of the augmented dataset. We sampled 1M of transitions among 5M generated transitions to effectively test these metrics.

### D.1   Dynamic Plausibility

Dynamic plausibility, or Dynamic Mean Squared Error (Dynamic MSE), assesses the congruence of generated subtrajectories with the environment dynamics, following methodologies from prior works [41, 13]. Let $\mathcal{D}_\text{A}$ represent the augmented dataset. $(s, a, r, s')$ denotes a single transition, and $f^*$ denotes the true dynamic model of the environment. Then, dynamic MSE is computed by the error between the generated next state $s'$ and the true next state $f^*(s, a)$. Prior to error computation, we normalize each state.

$$\text{Dynamic MSE}(\mathcal{D}_\text{A}) = \frac{1}{|\mathcal{D}_\text{A}|} \sum_{(s,a,r,s') \in \mathcal{D}_\text{A}} (f^*(s,a) - s')^2 \tag{14}$$

### D.2   Novelty

Novelty metric, inspired by [42, 37], quantifies the uniqueness of the augmented trajectories compared to the original offline data. Novelty is calculated as the average l2 distance between each generated trajectory and its nearest offline data point. To gain a deeper understanding of the generated data, we categorized novelty into state novelty, action novelty, and state-action novelty. The calculation formula is as follows:

$$\text{Novelty}(\mathcal{D}_\text{A}, \mathcal{D}) = \frac{1}{|\mathcal{D}_\text{A}|} \sum_{(s,a,r,s') \in \mathcal{D}_\text{A}} \min_{(\bar{s},\bar{a},\bar{r},\bar{s}') \in \mathcal{D}} ((s,a) - (\bar{s},\bar{a}))^2 \tag{15}$$

where $\mathcal{D}$ is offline dataset.

### D.3   Optimality

Lastly, optimality reflects the ability of generated trajectories to achieve high rewards. We quantify optimality as an oracle reward, which is computed as the average of the reward values $\tilde{r}$ obtained by querying the environment with the generated states and actions $(s, a)$. This metric serves as an indicator of the effectiveness of the generated trajectories in terms of reward maximization.

## E   Additional Experimental Results

In this section, we present additional experiment results to deepen our understanding on GTA.

### E.1   Ablation on trajectory-level generation

Table 10 shows that reducing the trajectory generation to a single transition significantly raises the Dynamic MSE and degrades policy performance. Once the generation target becomes trajectory, horizon length marginally impacts both performance and Dynamic MSE. These results emphasize the importance of leveraging sequential relationships within transitions when augmenting trajectories.

Table 10: Ablations on the length of horizon $H$ in halfcheetah-medium-v2

| metric | 1 | 16 | 32 | 64 | 128 |
|---|---|---|---|---|---|
| D4RL score | $46.59 \pm 0.24$ | $55.85 \pm 0.54$ | $57.41 \pm 0.53$ | $58.40 \pm 0.69$ | $56.93 \pm 0.89$ |
| Dynamic MSE ($\times 10^{-2}$) | 5.12 | 1.11 | 0.85 | 0.66 | 0.65 |

### E.2   Mixed-quality Dataset

It is widely known that the performance of offline RL significantly varies depending on the performance of the behavior policy used to collect the data [15]. However, offline RL algorithms often fail

to fully exploit the experience from high-performing trajectories [43] when the offline dataset is a mixed-quality dataset, which consists of few high-performing trajectories and many low-performing trajectories. Here, we demonstrate that GTA can effectively address mixed-quality dataset issues by augmenting high-performing trajectories using information from low-performing trajectories.

In this experiment, we mix the halfcheetah-medium-replay-v2 and halfcheetah-expert-v2 data with ratios of 1:20 and 1:10. We choose these two datasets as the performance gap between them is the largest among the Gym locomotion tasks. The halfcheetah-medium-replay-v2 dataset consists of 200k transitions, and the expert transitions in the mixed-quality dataset are 10k and 20k each.

We augment the mixed-quality dataset with various methods, including naive duplication, S4RL, SynthER, and GTA. Naive duplication is the setting where high-performing trajectories are duplicated to 5M transitions without any modification.

To prove that GTA can exploit information on low-performing trajectories for augmenting high-performing trajectories, we additionally trained TD3BC on the expert-only configuration where the dataset consists of only high-performing trajectories used for comprising mixed-quality datasets. We employ S4RL, SynthER, and GTA as an augmentation method for expert-only configuration. We hypothesize that If the result of the GTA on the mixed-quality outperforms GTA on expert-only, GTA exploits the information of low-performing trajectories while augmenting high-performing trajectories.

We train GTA on the mixed-quality or expert-only dataset and augment only the high-performing trajectories. Similarly, we train SynthER on both datasets and generate the new transitions from the diffusion model. By the *partial noising and denoising technique*, GTA can choose the specific data to augment, while SynthER cannot because SynthER starts generating the transition from the Gaussian noise without any guidance.

Table 11 demonstrates that GTA shows a significant performance boost on mixed-quality configurations. The result also exhibits that GTA on the mixed-quality configuration outperforms GTA on the expert-only configuration. From these results, we suggest that GTA can effectively augment the high-performing trajectories while exploiting the information on low-performing trajectories. Additionally, the exceptionally superior performance of GTA on expert-only configuration implies that GTA can enhance the sample efficiency considerably.

Table 11: D4RL score of the TD3BC on sparse expert dataset. The results are calculated with 10 evaluations over 8 seeds.

| Data Configuration | Aug. | Expert Ratio | |
| --- | --- | --- | --- |
| | | 1:20 | 1:10 |
| Baseline | None | 44.64 $\pm$ 0.71 | |
| Mixed-quality | None | 42.67 $\pm$ 4.44 | 44.47 $\pm$ 4.45 |
| | Naive Duplication | 29.23 $\pm$ 5.52 | 41.17 $\pm$ 12.26 |
| | S4RL | 38.43 $\pm$ 7.79 | 41.95 $\pm$ 4.91 |
| | SynthER | 2.00 $\pm$ 0.09 | 2.02 $\pm$ 0.05 |
| | GTA | **69.30 $\pm$ 6.79** | **59.54 $\pm$ 17.69** |
| Expert-only | None | -0.83 $\pm$ 1.12 | 1.82 $\pm$ 2.19 |
| | S4RL | -0.73 $\pm$ 1.92 | 1.84 $\pm$ 1.04 |
| | SynthER | -0.91 $\pm$ 0.83 | 2.31 $\pm$ 1.08 |
| | GTA | **36.26 $\pm$ 8.20** | **48.49 $\pm$ 3.81** |

## E.3 Effectiveness of GTA Under The Data Sparsity

To assess the effectiveness of GTA when the number of transitions in the offline data is small, we designed experiments that utilized only 5%, 10%, 15%, and 20% of the total offline data for the training diffusion model. We then evaluated GTA with TD3BC in the halfcheetah-medium-v2 environment. Table 12 demonstrate that Even with only 5% of the data, GTA outperforms the TD3BC baseline implemented with 100% of the data. This implies that GTA can efficiently improve the sample efficiency of offline RL algorithm when the volume of the offline dataset is limited.

Table 12: D4RL score of the TD3BC on small dataset. The results are calculated with 10 evaluations over 4 seeds.

| metric | 5% | 10% | 15% | 20% | 100% |
|---|---|---|---|---|---|
| TD3BC | 46.96 ± 0.34 | 48.28 ± 0.21 | 48.36 ± 0.25 | 48.33 ± 0.21 | 48.65±0.31 |
| TD3BC + GTA | 52.81 ± 0.29 | 53.49 ± 0.38 | 53.63 ± 1.92 | 55.39 ± 0.61 | 57.41 ± 0.53 |

## E.4 Effectiveness of GTA on sequence modeling-based approaches

Unlike prior works that augment data at the transition level, GTA generates trajectories and therefore can be applied to sequence modeling approaches such as Decision Transformer [DT, 38]. For training DT, we should assign returns-to-go token for each timestep $t$, which is calculated as $\hat{R}_t = \sum_{k=t}^{T} r_k$. To model return-to-go for our generated subtrajectory, we set the return-to-go of the first timestep of the subtrajectory as $\hat{R}_t$, where $\hat{R}_t$ can be computed from the original data. For subsequent timesteps, we deduct the reward that we generated.

$$\boldsymbol{\tau} = \begin{bmatrix} s_t & s_{t+1} & \cdots & s_{t+H-1} \\ a_t & a_{t+1} & \cdots & a_{t+H-1} \\ r_t & r_{t+1} & \cdots & r_{t+H-1} \end{bmatrix} \longrightarrow \boldsymbol{\tau'} = \begin{bmatrix} s'_t & s'_{t+1} & \cdots & s'_{t+H-1} \\ a'_t & a'_{t+1} & \cdots & a'_{t+H-1} \\ r'_t & r'_{t+1} & \cdots & r'_{t+H-1} \end{bmatrix} \quad (16)$$

$$\hat{\boldsymbol{R}}' = \begin{bmatrix} \hat{R}_t & \hat{R}_t - r'_t & \cdots & \hat{R}_t - \sum_{k=t}^{t+H-2} r'_k \end{bmatrix} \quad (17)$$

To test the efficacy of GTA on sequence modeling, we train a DT on Gym locomotion tasks and Maze2d environments. The result in Table 13 demonstrates that DT with GTA outperforms baselines in terms of average performance at Gym locomotion tasks. And Table 14 shows that DT with GTA augmented data exhibits superior performance with respect to average score. These results highlight the effectiveness of trajectory-level data augmentation in preserving sequential relationships and leveraging the learned long-term dynamics.

Table 13: Normalized average scores on D4RL MuJoCo locomotion tasks, with the highest scores highlighted in **bold**. The results are calculated with from 100 evaluations using 4 seeds.

| Algo. | Aug. | Halfcheetah | | | Hopper | | | Walker2d | | | Average |
|---|---|---|---|---|---|---|---|---|---|---|---|
| | | medium | medium-replay | medium-expert | medium | medium-replay | medium-expert | medium | medium-replay | medium-expert | |
| DT | None | 42.43 ± 0.14 | **39.34 ± 1.22** | **92.43 ± 0.50** | 63.09 ± 2.49 | **81.81 ± 3.39** | 109.05 ± 2.02 | 71.64 ± 0.69 | 62.06 ± 1.88 | 108.38 ± 0.31 | 74.47 ± 1.40 |
| | S4RL | 42.44 ± 0.39 | 38.71 ± 0.83 | 91.80 ± 0.77 | 64.49 ± 1.70 | 66.47 ± 19.27 | **110.57 ± 0.81** | 72.22 ± 2.49 | 59.67 ± 4.91 | **108.40 ± 0.24** | 72.75 ± 3.49 |
| | GTA | **43.83 ± 0.13** | 37.98 ± 4.97 | 91.78 ± 1.88 | **64.57 ± 1.22** | 78.43 ± 9.93 | 110.54 ± 0.18 | **74.94 ± 1.72** | **67.90 ± 13.41** | 108.24 ± 0.62 | **75.36 ± 3.78** |

Table 14: Normalized evaluation scores of Decision Transformer on D4RL Maze 2d tasks. The results are calculated with 100 evaluations over 4 seeds.

| Algo. | Aug. | maze2d | | | Average |
|---|---|---|---|---|---|
| | | umaze | medium | large | |
| DT | None | 39.43 ± 13.49 | **55.24 ± 32.82** | 18.49 ± 24.14 | 37.72 ± 23.15 |
| | S4RL | 39.38 ± 13.51 | 55.05 ± 31.79 | 18.69 ± 24.46 | 37.71 ± 23.25 |
| | GTA | **50.19 ± 24.37** | 49.60 ± 5.59 | **25.49 ± 25.79** | **41.76 ± 18.58** |

## E.5 Ablation on Reweighted Sampling

We conduct an ablation study on reweighted sampling, which is additionally introduced to make the diffusion model focus on high-reward regions. We conduct experiments on two environments (Hopper-medium and Walker2d-medium) to verify the effectiveness of the reweighted sampling. Table 15 presents evaluation scores and oracle rewards with and without reweighted sampling. As shown in the table, reweighted sampling achieves a higher D4RL score compared to its counterpart, showcasing its effectiveness. We also observe that the oracle reward of the generated dataset is also increased, indicating that our reweighted sampling strategy successfully positions more points in higher reward regions.

## E.6 Ablation on Different Conditioning Strategies

We explored alternative design choices for GTA that exploit the inpainting ability of the diffusion model. GTA condition on amplified return value to improve the optimality of the offline dataset.

Table 15: Ablations on the reweighted sampling technqiues

| Env | D4RL score | | Oracle reward | |
|---|---|---|---|---|
| | O | X | O | X |
| Hopper-medium-v2 | $57.45 \pm 4.54$ | $56.95 \pm 6.30$ | 3.61 | 3.60 |
| Walker2d-medium-v2 | $88.26 \pm 0.80$ | $86.43 \pm 4.60$ | 4.35 | 4.17 |

However, another design choice is to condition the diffusion model by inpainting each step of reward with amplified reward. We named this method as amplified reward inpainting. We implement reward inpainting by multiplying the reward of each trajectory by a multiplier $\alpha = 1.3$ and continually replacing generated rewards with multiplied rewards of the original trajectory during the denoising process of the diffusion model. Note that we do not apply the return conditioning originally used in GTA when amplified reward inpainting is applied. Table 16 demonstrates that the trajectories augmented with reward inpainting yield lower returns than GTA, especially a significant performance drop on the Hopper-medium dataset.

Table 16: D4RL score comparison between two conditioning strategies, amplified return guidance of GTA and amplified reward inpainting

| strategy | Halfcheetah-medium | Hopper-medium | Walker2d-medium |
|---|---|---|---|
| amplified return guidance (GTA) | $57.41 \pm 0.53$ | $60.34 \pm 3.27$ | $87.07 \pm 0.45$ |
| amplified reward inpainting | $55.13 \pm 0.80$ | $50.68 \pm 5.29$ | $87.30 \pm 1.25$ |

Additionally, we conducted a comparative analysis of the augmented trajectories with different conditioning strategies, examining both the dynamic MSE and the oracle reward. As illustrated in Table 17, the dynamics MSE for data augmented through reward inpainting is significantly higher compared to the one augmented with GTA. Oracle reward of data augmented through reward inpainting is also lower than GTA. We hypothesize that reward inpainting introduces constraints that reduce the flexibility of states and actions, resulting in poor dynamic MSE and degrading the underlying offline RL policy.

Table 17: Data quality comparison between two conditioning strategies, amplified return guidance of GTA and amplified reward inpainting

| strategy | Dynamic MSE ($\downarrow$) ($\times 10^{-2}$) | Oracle Reward ($\uparrow$) |
|---|---|---|
| amplified return guidance (GTA) | 0.92 | 6.54 |
| amplified reward inpainting | 8.51 | 6.26 |

### E.7 Ablation on utilizing reward proxy model

Instead of using augmented reward, we additionally train the reward model to replace it. Table 18 compares performance using a reward function instead of augmented rewards.

Although the method that replaces augmented rewards with the predictions of the proxy reward model shows comparable performances, incorporating the reward model into the augmentation process would additionally involve a reward labeling step for augmented states and actions before training the offline RL algorithm. Therefore, we opted to generate all states, actions, and rewards at once to achieve a more straightforward yet more effective augmentation.

Table 18: Normalized return on halfcheetah-medium-v2 with TD3BC.

| metric | Reward proxy | GTA |
|---|---|---|
| halfcheetah-medium | $57.96 \pm 0.49$ | $57.69 \pm 0.73$ |

# F Additional Analysis

In this section, we present additional analysis that are not included in the main section due to the page limit.

## F.1 Comparing Conditioning Strategies

A common approach for providing classifier-free guidance is to condition on a fixed value, representing the desired return according to the environment [22, 14]. However, this strategy can lead to significant issues when jointly implemented with *partial noising*. In the scenario where a trajectory is denoised from the partially noised trajectory, conditioning on fixed return, which is excessively greater than the return of the original trajectory, collides with the pre-existing context.

When the partially noised trajectory is denoised with guidance towards the fixed excessively high return value, the guidance collides with the pre-existing context. Such forceful conditioning can lead to the generation of invalid trajectories. Furthermore, even without applying partial noising and denoising, using a fixed value tends to decrease the diversity of the samples because of the narrow extent of the conditioning value.

To this end, we analyze the fixed conditioning method in more detail. Table 19 shows the quality measurements of the dataset generated with a fixed conditioning. We observe that fixed-value conditioning is capable of generating novel and high-rewarding samples. However, the dynamic plausibility significantly increases due to the forceful conditioning.

Table 19: Data quality comparison on halfcheetah-medium-v2 between distinct conditioning strategies: fixed conditioning, unconditioning, and amplified return conditioning.

| metric | Baseline | Fixed | Unconditioned | Amplified (ours) |
|---|---|---|---|---|
| Dynamic MSE ($\times 10^{-2}$) | 0.00 | 4.88 | 0.91 | 0.85 |
| Oracle reward | 4.77 | 4.89 | 4.84 | 6.08 |
| Novelty | - | 1.41 | 1.42 | 1.58 |

## F.2 Statistical Test Results for The Performance Boosting Effects of GTA

We investigate the statistical meaningfulness of the performance improvement compared to the ones of the SynthER. To compute the standard deviation for the average performance of each offline RL algorithm, we first organize the final performance by seed and calculate the mean score of each seed across the tasks of the table. Next, we calculate the standard deviation of these mean scores for each seed and report this value as the standard deviation of the algorithm's performance. This protocol for calculating standard deviation is applied at Table 1 and Table 2.

Based on the t-test results shown in Table 20, we suggest that GTA induces a statistically significant performance boost over SynthER, achieving p-values below 0.05 across all algorithms. The p-value result on Table 20 is calculated by the Welch's t-test for t-statistic [44] and the experiment results are sourced from the Table 1.

Table 20: Welch's t-test result demonstrates statistically significant performance improvements of GTA over SynthER

| | TD3BC | IQL | CQL | MCQ |
|---|---|---|---|---|
| SynthER | $78.64 \pm 2.38$ | $81.20 \pm 1.46$ | $81.97 \pm 2.21$ | $82.81 \pm 3.21$ |
| GTA | $84.63 \pm 2.20$ | $85.27 \pm 1.02$ | $86.11 \pm 0.94$ | $85.91 \pm 1.05$ |
| $p$-value | 0.0001 | 0.0000 | 0.0008 | 0.0303 |

## F.3 Impact of Partial Noising and Denoising Framework on Terminal States

We further explore the effects of *partial noising and denoising framework*, particularly focusing on how well they preserve original trajectory information, especially terminal states. Halfcheetah contrasts with hopper and walker2d environments in terms of the existence of episode termination.

While hopper and walker2d have terminals, the sparsity of terminal states in datasets, exemplified by their $0.0001\%$ occurrence in walker2d transitions, underscores the challenge of preserving this critical information.

Our findings, illustrated in Figure 6, show that in Walker2d, a high noise level ($\mu = 1$) leads to unstable learning due to excessive deviation from the original trajectory. In contrast, a lower noise level ($\mu = 0.5$) maintains more of the original trajectory data, resulting in more stable learning outcomes. This highlights the importance of an approach to the balance of retaining original information in environments with terminal states.

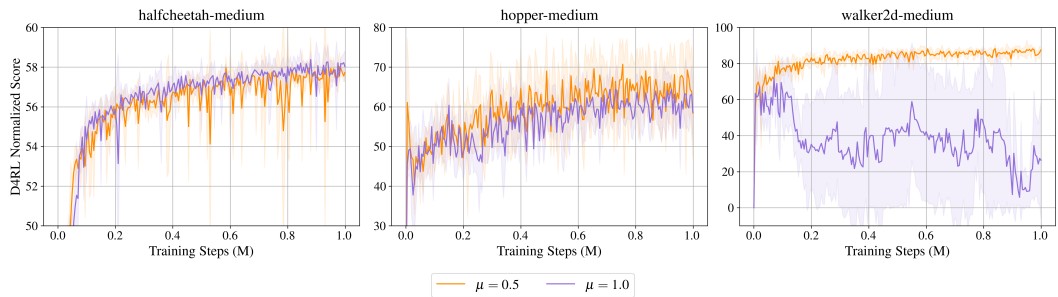

Figure 6: Normalized average scores on D4RL locomotion environments. Walker2d underperforms when the noise level is set to $\mu = 1$, leading to the loss of crucial terminal information

## F.4  Data Quality Table for the Figure 5

In this section, we provide the exact results in addition to the analysis of the data quality metric presented in Section 5.4. Table 21, Table 22, and Table 23 exhibit the optimality, novelty, and dynamic MSE of GTA and baseline augmentation methods. For a clear presentation of Figure 5, we take the logarithm of the Dynamic MSE. For novelty and oracle reward, we adjust the axis scale for each environment.

As shown in Table 21 and Table 22, GTA consistently achieves higher values than other augmentation methods at both the oracle reward and novelty across all locomotion environments. Simultaneously, the dynamic MSE remains comparably small. These results suggest that GTA successfully generates data that is novel, high-rewarding, and dynamically plausible.

Beyond comparing the novelty of state-action pairs, we measured the novelty by separating state and action to assess the ability of GTA to discover novel states as well as novel actions. As seen in Table 25 and Table 24, GTA consistently outperforms other methods in generating novel states and actions.

Table 21: Comparison of oracle rewards across gym locomotion tasks.

| Optimality ($\uparrow$) | S4RL | Synther | GTA |
|---|---|---|---|
| Halfcheetah-medium-v2 | 4.77 | 4.78 | 6.08 |
| Halfcheetah-medium-replay-v2 | 3.10 | 3.09 | 3.99 |
| Halfcheetah-medium-expert-v2 | 7.73 | 7.7 | 10.07 |
| Hopper-medium-v2 | 3.11 | 3.10 | 3.28 |
| Hopper-medium-replay-v2 | 2.38 | 2.37 | 2.70 |
| Hopper-medium-expert-v2 | 3.36 | 3.35 | 3.87 |
| Walker2d-medium-v2 | 3.39 | 3.40 | 3.51 |
| Walker2d-medium-replay-v2 | 2.47 | 2.46 | 3.35 |
| Walker2d-medium-expert-v2 | 4.15 | 4.17 | 4.81 |

Table 22: Comparison of the novelty of state and action across gym locomotion tasks.

| Novelty (↑) | S4RL | Synther | GTA |
|---|---|---|---|
| Halfcheetah-medium-v2 | 1.39 | 1.41 | 1.58 |
| Halfcheetah-medium-replay-v2 | 1.78 | 1.88 | 1.91 |
| Halfcheetah-medium-expert-v2 | 1.22 | 1.22 | 1.3 |
| Hopper-medium-v2 | 1.36 | 1.37 | 2.05 |
| Hopper-medium-replay-v2 | 2.03 | 2.10 | 2.28 |
| Hopper-medium-expert-v2 | 1.45 | 1.46 | 1.79 |
| Walker2d-medium-v2 | 1.26 | 1.26 | 1.31 |
| Walker2d-medium-replay-v2 | 1.84 | 1.93 | 2.00 |
| Walker2d-medium-expert-v2 | 1.16 | 1.18 | 1.30 |

Table 23: Comparison of Dynamic MSE across gym locomotion tasks.

| Dynamic MSE (↓) ($\times 10^{-2}$) | S4RL | Synther | GTA |
|---|---|---|---|
| Halfcheetah-medium-v2 | 0.0 | 0.73 | 0.85 |
| Halfcheetah-medium-replay-v2 | 0.01 | 1.58 | 1.25 |
| Halfcheetah-medium-expert-v2 | 0.0 | 0.91 | 0.79 |
| Hopper-medium-v2 | 0.07 | 1.56 | 0.08 |
| Hopper-medium-replay-v2 | 0.06 | 1.35 | 0.12 |
| Hopper-medium-expert-v2 | 0.04 | 0.76 | 0.08 |
| Walker2d-medium-v2 | 1.90 | 2.09 | 2.98 |
| Walker2d-medium-replay-v2 | 2.59 | 3.37 | 2.77 |
| Walker2d-medium-expert-v2 | 1.63 | 1.73 | 3.49 |

Table 24: Comparison of the novelty of state across gym locomotion tasks.

| Novelty (↑) | S4RL | Synther | GTA |
|---|---|---|---|
| Halfcheetah-medium-v2 | 1.12 | 1.14 | 1.28 |
| Halfcheetah-medium-replay-v2 | 1.36 | 1.44 | 1.47 |
| Halfcheetah-medium-expert-v2 | 0.95 | 0.94 | 1.01 |
| Hopper-medium-v2 | 0.38 | 0.39 | 0.69 |
| Hopper-medium-replay-v2 | 0.63 | 0.66 | 0.74 |
| Hopper-medium-expert-v2 | 0.41 | 0.41 | 0.54 |
| Walker2d-medium-v2 | 0.81 | 0.81 | 0.85 |
| Walker2d-medium-replay-v2 | 1.21 | 1.28 | 1.35 |
| Walker2d-medium-expert-v2 | 0.71 | 0.73 | 0.81 |

Table 25: Comparison of the novelty of action across gym locomotion tasks.

| Novelty (↑) | S4RL | Synther | GTA |
|---|---|---|---|
| Halfcheetah-medium-v2 | 0.22 | 0.22 | 0.23 |
| Halfcheetah-medium-replay-v2 | 0.28 | 0.29 | 0.3 |
| Halfcheetah-medium-expert-v2 | 0.24 | 0.23 | 0.25 |
| Hopper-medium-v2 | 0.07 | 0.07 | 0.08 |
| Hopper-medium-replay-v2 | 0.07 | 0.07 | 0.08 |
| Hopper-medium-expert-v2 | 0.07 | 0.07 | 0.08 |
| Walker2d-medium-v2 | 0.35 | 0.35 | 0.36 |
| Walker2d-medium-replay-v2 | 0.41 | 0.42 | 0.44 |
| Walker2d-medium-expert-v2 | 0.36 | 0.36 | 0.38 |

## F.5 Comparisons of GTA with Offline Model-based RL and Diffusion Planners

In this section, we compare the performance and computational cost of GTA with offline model-based RL methods and diffusion planners. While GTA shares commonalities with both approaches as discussed in Section 2, it introduces key approaches that lead to improved performance and test-time computation efficiency.

Table 26 demonstrates superior performance of GTA compared to both model-based RL algorithms and diffusion planners in locomotion tasks. The results suggest that trajectory augmentation via diffusion models can be more effective than traditional single-transition dynamic models used in offline model-based RL. Additionally, GTA outperforms diffusion planners, which require a time-intensive sampling process during decision-making.

We also analyze the computational efficiency of GTA compared to baseline methods. As shown in Table 27, GTA exhibits significant computational advantages over diffusion planners during test-time evaluation. This advantage stems from shifting the computational burden from the sampling phase to the data preparation stage.

The model training process for both GTA and offline model-based RL algorithms [25, 27] consists of two stages: dynamic model training and policy training. GTA takes longer for dynamic model training due to its diffusion model with millions of parameters. However, offline model-based RL algorithms require significantly more time for policy training because model rollouts slow down this step. Overall, the total training time for COMBO and GTA is comparable, but GTA demonstrates superior performance, as shown in Table 26.

Table 26: Comparison with diffusion planners and offline model-based RL baselines with GTA. For baselines, we take results from their original papers. For GTA, we report score of TD3BC and CQL as base algorithms.

| Dataset Type | Diffusion Planners | | | Model-based RL | | | GTA | |
|---|---|---|---|---|---|---|---|---|
| | Diffuser | DD | AdaptDiffuser | MOPO | MOReL | COMBO | TD3BC | CQL |
| halfcheetah-medium | 42.8 | 49.1 | 44.2 | 42.3 | 42.1 | 54.2 | **57.84 ± 0.51** | 54.14 ± 0.31 |
| halfcheetah-medium-replay | 37.7 | 39.3 | 38.3 | 53.1 | 40.2 | **55.1** | 50.04 ± 0.84 | 51.36 ± 0.27 |
| halfcheetah-medium-expert | 88.9 | 90.6 | 89.6 | 63.3 | 53.3 | 90.0 | 93.13 ± 3.07 | **94.93 ± 3.71** |
| hopper-medium | 74.3 | 79.3 | 92.2 | 28.0 | 95.4 | **97.2** | 69.57 ± 4.05 | 74.80 ± 7.42 |
| hopper-medium-replay | 93.6 | **100.0** | 96.6 | 67.5 | 93.6 | 89.5 | 89.31 ± 16.84 | 98.88 ± 3.51 |
| hopper-medium-expert | 103.3 1 | **111.8** | 111.6 | 23.7 | 108.7 | 111.1 | 110.40 ± 4.04 | 110.90 ± 3.44 |
| walker2d-medium | 79.6 | 82.5 | 84.7 | 17.8 | 77.8 | 81.9 | **86.69 ± 0.89** | 80.40 ± 4.98 |
| walker2d-medium-replay | 70.6 | 75.0 | 84.4 | 39.0 | 49.8 | 56.0 | **93.82 ± 1.74** | 91.57 ± 5.15 |
| walker2d-medium-expert | 106.9 | 108.8 | 108.2 | 44.6 | 95.6 | 103.3 | **110.86 ± 0.34** | 110.44 ± 0.28 |
| locomotion average | 77.5 | 81.8 | 83.4 | 42.60 | 72.72 | 82.03 | 84.63 ± 2.20 | **85.27 ± 1.02** |

Table 27: Comparison of computational cost between Diffusion planners, offline model-based RL baselines, and GTA. The elapsed time was measured using a single RTX 3090 on the HalfCheetah-medium-v2 environment.

| | Diffusion Planners | | Model-based RL | | GTA |
|---|---|---|---|---|---|
| | DD | AdaptDiffuser | MOPO | COMBO | TD3BC |
| Model Training (H) | 32.0 | 45.3 | 1.7 | 1.7 | 53.0 |
| RL Policy Training (H) | - | - | 20.8 | 45.8 | 2.0 |
| Synthetic Data Generation (H) | - | 2.4 | - | - | 3.5 |
| Policy Evaluation (s) | 1.3 | 1.4 | $6 \times 10^{-4}$ | $7 \times 10^{-4}$ | $3 \times 10^{-4}$ |

## F.6 Correlation on Conditioning Value and Generated Reward

To evaluate the controllability of GTA in trajectory generation, we analyze the Pearson correlation between the sum of rewards for generated subtrajectories and the conditioning value. As shown in Figure 7, the correlation coefficients of Halfcheetah medium, Halfcheetah medium-replay and Halfcheetah medium-expert datasets are 0.55, 0.91, and 0.99, respectively. These results demonstrate that our diffusion model can effectively generate trajectories that align with the conditioned return values.

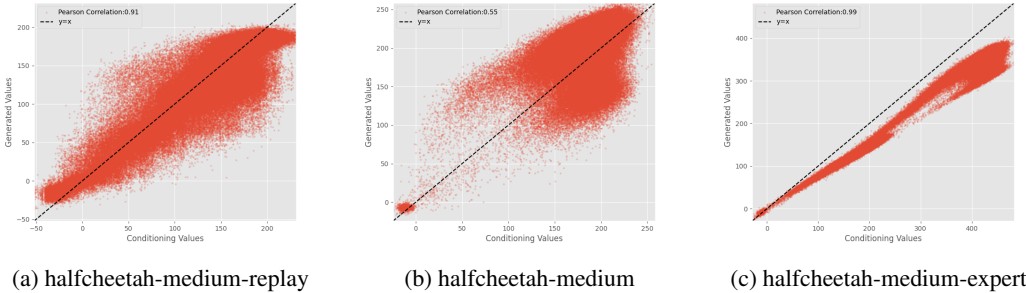

| (a) halfcheetah-medium-replay | (b) halfcheetah-medium | (c) halfcheetah-medium-expert |

Figure 7: Correlation on conditioning value and generated reward

## F.7 Oracle Reward Distribution of Generated Trajectories

We visualize the reward of the original dataset, generated reward from GTA, and real reward computed from the environment using state and action generated from GTA to visualize that GTA can really shift the data distribution to high reward region while preserving environmental dynamics. We present the results conducted on Halfcheetah-medium-v2 in the main section. Other results are presented below. As shown in the figures, GTA mostly generates valid and high-reward trajectories across various environments and datasets.

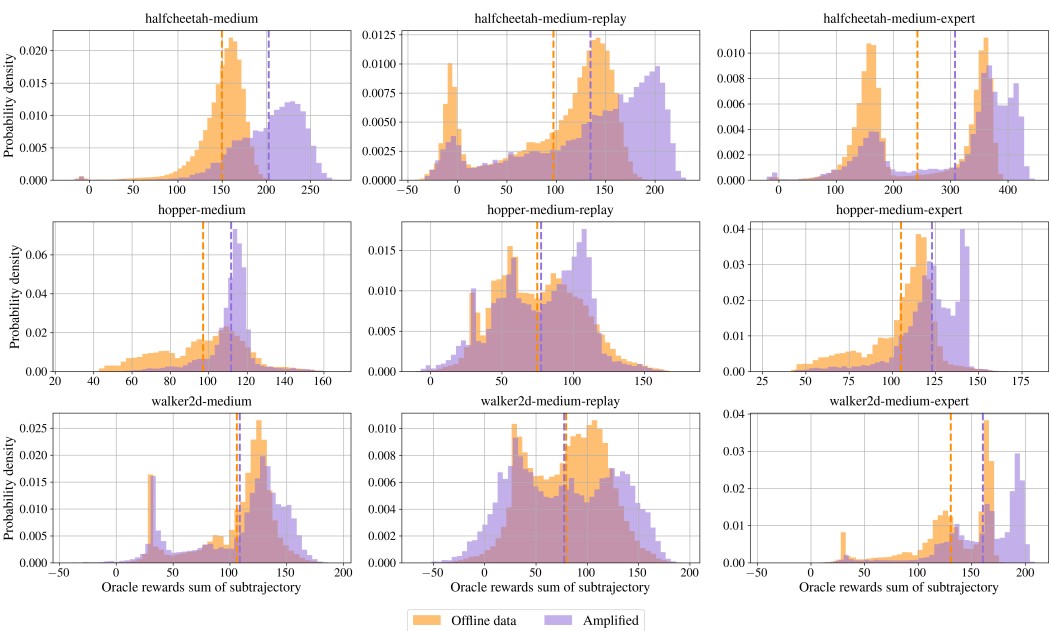

Figure 8: Oracle reward sum of subtrajectories on D4RL Gym locomotion

## G    GTA Sensitivity Test

In this section, we conduct a sensitivity test on hyperparameters introduced in GTA. To demonstrate the sensitivity of GTA on these hyperparameters, we choose Halfcheetah-medium-v2 dataset with TD3+BC as a baseline Offline RL algorithm. For all experiments, we train the algorithm with 1M gradient steps with 4 random seeds.

### G.1    Batch Size and Training Epochs for Offline RL Training

We further verify that increasing batch size and training epochs affect the performance of our method. Most prior works fix batch size as 256 and training epochs as 1M, which is a reasonable choice where

the original dataset size is quite small (100K to 2M). However, as we upsample the dataset into 5M, it is natural to check whether increasing batch size or training epochs might lead to an improvement in the performance. To validate this, we perform experiments with the upsampled dataset by varying batch size and training epochs.

Figure 9 shows the performance of the TD3BC algorithm trained with an augmented dataset from GTA by varying batch size. We observe that increasing batch size significantly improves the performance. Table 28 illustrates the performance of offline RL algorithms trained with the original offline dataset. As illustrated in the table, there is no big difference in performance when we just naively scale up the batch size. We attribute this to the natural phenomenon of neural networks to generalize well with a larger batch size if there is more data available for training. To this end, we set the batch size as 1024 for all methods in the main experiments.

As shown in the Figure 10, we observe that as we increase the training epochs, GTA can achieve more performance gain. We also observe that using both large batch size and more training epochs continually improves the performance. After 10M training steps with a batch size of 1024, we achieve significant improvement compared to the TD3BC trained with the original dataset, trained 1M steps with a batch size of 1024.

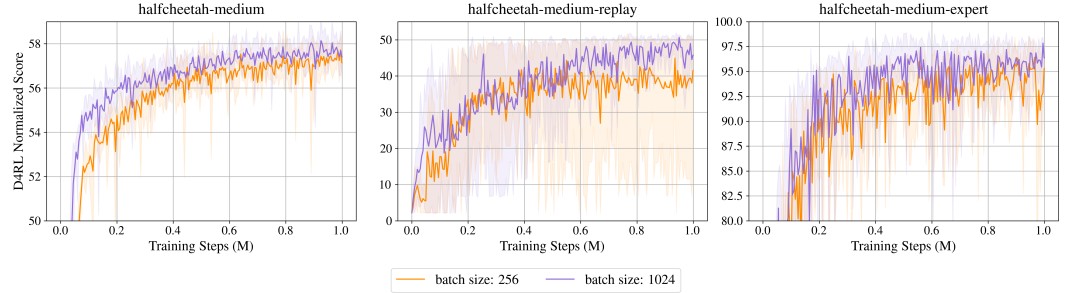

Figure 9: Performance of TD3BC on Halfcheetah environments with different batch sizes. Training with a larger batch size achieves higher scores.

Table 28: Normalized average scores on D4RL MuJoCo locomotion tasks. Training with a larger batch size makes no difference without modification on the offline dataset.

| Algorithm | Batch Size | Halfcheetah | | | Hopper | | | Walker2d | | | Average |
| --- | --- | --- | --- | --- | --- | --- | --- | --- | --- | --- | --- |
| | | medium | medium-replay | medium-expert | medium | medium-replay | medium-expert | medium | medium-replay | medium-expert | |
| TD3BC | 256 | $48.10 \pm 0.18$ | $44.84 \pm 0.59$ | $90.78 \pm 2.45$ | $60.37 \pm 3.49$ | $64.42 \pm 21.51$ | $101.17 \pm 9.07$ | $82.71 \pm 4.78$ | $85.62 \pm 4.01$ | $110.03 \pm 0.36$ | $76.45 \pm 5.56$ |
| | 1024 | $48.65 \pm 0.31$ | $44.77 \pm 0.70$ | $92.82 \pm 2.48$ | $61.89 \pm 3.38$ | $69.49 \pm 21.43$ | $105.91 \pm 4.63$ | $85.05 \pm 1.03$ | $85.88 \pm 2.89$ | $110.14 \pm 0.46$ | $78.29 \pm 4.15$ |
| CQL | 256 | $47.04 \pm 0.22$ | $45.04 \pm 0.27$ | $95.63 \pm 0.42$ | $59.08 \pm 3.77$ | $95.11 \pm 5.27$ | $99.26 \pm 10.01$ | $80.75 \pm 3.28$ | $73.09 \pm 13.22$ | $109.56 \pm 0.39$ | $78.28 \pm 4.19$ |
| | 1024 | $46.98 \pm 0.24$ | $44.37 \pm 0.25$ | $95.81 \pm 0.77$ | $62.40 \pm 3.69$ | $74.27 \pm 17.14$ | $106.08 \pm 8.01$ | $82.56 \pm 0.58$ | $79.06 \pm 6.47$ | $109.53 \pm 0.27$ | $77.90 \pm 4.16$ |
| IQL | 256 | $48.31 \pm 0.22$ | $44.46 \pm 0.22$ | $94.74 \pm 0.52$ | $67.53 \pm 3.78$ | $97.43 \pm 6.39$ | $107.42 \pm 7.80$ | $80.91 \pm 3.17$ | $82.15 \pm 3.03$ | $111.72 \pm 0.86$ | $81.63 \pm 2.89$ |
| | 1024 | $48.73 \pm 0.10$ | $43.50 \pm 0.46$ | $94.09 \pm 2.71$ | $64.75 \pm 8.09$ | $95.65 \pm 4.15$ | $102.02 \pm 10.59$ | $81.56 \pm 2.20$ | $71.93 \pm 13.13$ | $112.60 \pm 0.58$ | $79.43 \pm 4.67$ |

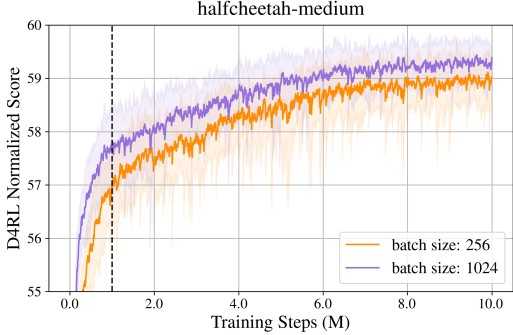

Figure 10: Ablations on training with more epochs. Training more epochs leads to improvement in performance.

## G.2 Scale of Multiplier $\alpha$

The scale of multiplier $\alpha$ determines the strength of model exploitation towards the high reward region. If we set $\alpha$ high, GTA tries to generate high-reward trajectories compared to the original trajectories, which might lead to generating dynamically infeasible trajectories. If $\alpha$ leans towards 1, it would generate trajectories with similar return values to original trajectories and cannot push data distribution towards high-reward regions. Table 29 shows the performance of TD3+BC trained with generated data across different $\alpha$ values. We observe that there is a significant drop in performance where $\alpha$ is too high (e.g., 2.0). Also, we find that $1.1 \leq \alpha \leq 1.5$ generally exhibits good performance.

## G.3 Scale of Partial Noising $\mu$

The scale of partial noising ratio $\mu$ controls the balance between generating in-distribution and OOD trajectories. If we set $\mu \approx 0$, GTA generates trajectories similar to the original dataset and stays close to the original data distribution. If we set $\mu \approx 1$, it can generate novel trajectories via exploration while possessing the risk of violating environment dynamics. Intuitively, the high value of $\mu$ might lead to a significant performance drop due to the over-exploration of unknown regions. However, we observe that even when we choose relatively high $\mu$ (even set $\mu = 1$), it occasionally generates not only novel but also dynamically plausible trajectories. Table 29 shows the D4RL score by varying $\mu$. The table shows that data generated with relatively high $\mu$ exhibits higher performance. However, the aforementioned trend is not always true for all environments, as we have already discussed in Section 5.3.

Table 29: Sensitivity test on partial noising level $\mu$ and guidance multiplier $\alpha$

| $\mu \backslash \alpha$ | $\times 1.1$ | $\times 1.2$ | $\times 1.3$ | $\times 1.4$ | $\times 1.5$ | $\times 2.0$ |
|---|---|---|---|---|---|---|
| 0.1 | $48.56 \pm 0.44$ | $48.43 \pm 0.35$ | $48.54 \pm 0.51$ | $48.69 \pm 0.35$ | $48.57 \pm 0.20$ | $48.85 \pm 0.41$ |
| 0.25 | $48.62 \pm 0.29$ | $48.97 \pm 0.3$ | $49.48 \pm 0.53$ | $50.21 \pm 0.3$ | $51.30 \pm 0.49$ | $53.32 \pm 0.85$ |
| 0.5 | $53.55 \pm 0.82$ | $57.08 \pm 0.34$ | $57.94 \pm 0.49$ | $58.01 \pm 0.42$ | $57.80 \pm 0.53$ | $53.39 \pm 0.65$ |
| 0.75 | $54.39 \pm 0.57$ | $56.89 \pm 0.53$ | $57.85 \pm 0.27$ | $58.14 \pm 0.59$ | $58.09 \pm 0.29$ | $54.19 \pm 0.52$ |
| 1.0 | $54.42 \pm 0.13$ | $57.08 \pm 0.61$ | $56.58 \pm 2.84$ | $58.21 \pm 0.22$ | $58.02 \pm 0.34$ | $53.84 \pm 0.80$ |

## G.4 Size of Augmented Dataset

In our main experiment section, we generate 5M transitions with the conditional diffusion model. We investigate the performance of our method in terms of the size of the upsampled dataset. To this end, we choose four different levels of size from the range [0.1M, 10M]. As shown in Figure 11, the performance gain achieved by the upsampled dataset increases as the size of the augmented dataset becomes larger and converges after 5M samples.

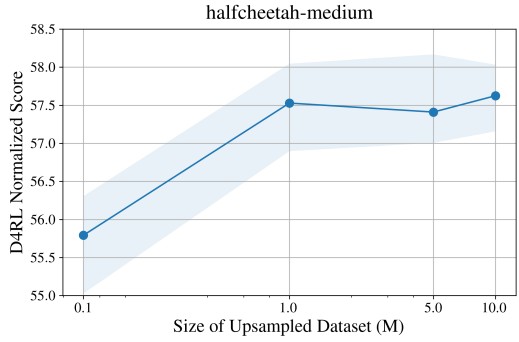

Figure 11: Ablation on the size of the upsampled dataset.

# H  Extended Related Works

## H.1  Related Works in Offline RL

In this section, we discuss related works with our work but just briefly discuss them in the main paper to clearly define the position of our work in offline RL literature.

- **BooT**: BooT [45] suggests a novel algorithm to train a Transformer-based model for offline RL by incorporating the idea of bootstrapping. BooT is a bootstrapping-based methodology that builds upon the Trajectory Transformer, reusing self-generated trajectories within the sequence model itself.

- **AdaptDiffuser**: AdaptDiffuser [24] is a planning method with a diffusion model that can self-evolve by training with self-generated diverse synthetic data using guidance from reward gradients. The AdaptDiffuser generates trajectories guided by various reward functions and trains the diffusion planner itself, shifting the data distribution learned by the diffusion model towards high reward regions.

**Main differences**: Our focus is building a high-quality augmented dataset that can be used for training any offline RL algorithms in a plug-and-play manner. We develop an algorithm-agnostic method to boost the overall performance of offline RL, not confined to specific kinds of model. Unlike our work, BooT and AdaptDiffuser utilize generated trajectories only for training itself and thereby are categorized as planners, not data augmentation, which is the reason why they are not included as baselines of our approach. While our method can also utilize bootstrapping, i.e., further training diffusion model with generated trajectories, we leave it as a future work.

## H.2  Related Works in Diffusion Models

We also notice that there are several approaches that resemble our approach, denoising from an intermediate state instead of pure random noise to preserve originality in various domains.

- **DiffAb**: DiffAb [46] use diffusion model to optimize existing antibodies by first perturbing the CDR sequence and denoise it. They observe that optimizing antibodies with this procedure can improve binding energy while keeping the sequence similar to the original one.

- **SDEdit**: SDEdit [47] highlights the key challenge of image editing as balancing the faithfulness and realism of synthetic images and proposes a methodology that leverages diffusion models to augment images. To enhance the realism of the generated images, it introduces partial noising and denoising, similar to the approach in GTA, where the editing is guided by the input image itself.

- **DA-Fusion**: DA-Fusion [48] conducted data augmentation utilizing pretrained diffusion model. They generate synthetic images using the partially noised images and then denoise them with guidance toward the original image. DA-Fusion significantly improved few-shot classification performance on the Pascal and COCO datasets.

# I  Broader Impacts

This paper is for the advancement of offline Reinforcement Learning. While it opens up various potential applications, any specific societal impacts are not immediately apparent and require further exploration.

