# OpenReview forum: "GTA: Generative Trajectory Augmentation with Guidance for Offline Reinforcement Learning"
_NeurIPS.cc/2024/Conference — NeurIPS 2024 poster_

### Official Review · Reviewer_UiuM · 2024-06-28

**Soundness:** 3
**Presentation:** 3
**Contribution:** 2
**Rating:** 6
**Confidence:** 4

**Summary:**

This paper tackles the challenges of Offline RL, which involves learning effective decision-making policies from static datasets without online interactions. The authors introduce Generative Trajectory Augmentation (GTA), a novel approach that uses a diffusion model to enhance offline data by augmenting trajectories to be both high-rewarding and dynamically plausible. GTA partially noises original trajectories and denoises them with classifier-free guidance based on amplified return values. The experiment results indicate that GTA, as a general data augmentation strategy, enhances the performance of widely used offline RL algorithms in both dense and sparse reward settings.

**Strengths:**

(1) Interesting and important topic: The guided data augmentation for offline RL is interesting. The success in this domain will further benefit the real-world applications of offline RL algorithms.

(2) Extensive experiments: the authors provide extensive experiments to validate their proposed method.

**Weaknesses:**

(1) Lack of theoretical guarantee to support the algorithm: The authors only provide empirical results, lacking theoretical analysis of the performance of the proposed method.

For other potential weaknesses, please see my questions.

**Questions:**

I will adjust my rating score based on the rebuttal results.

(1) How to select the hyperparameter \alpha in the denoising with amplified return guidance in equation (8). Can you also provide ablation study results for different selections of \alpha?

(2) In the generation phase, do you generate whole trajectories or generate subsequences of trajectories?

(3) Is the proposed method sensitive to the hyperparameter \mu? Based on the description in Figure 4 and Appendix F.3, when \mu = 1, it only performs well on the dataset with low quality. For high-quality datasets with abundant expert demonstration, does your method have trouble preserving and reconstructing the high-rewarding trajectories?

(4) Can you provide comparison results with diffusion-based offline RL methods? For example, Decision Diffuser [1] and AdapDiffuser [2].

(5) In the ablation study on reweighted sampling, what base offline RL do you use for D4RL score calculation?

(6) Since the reweighted sampling can implicitly change the dataset distribution, can you also apply the reweighted sampling technique as well for other baseline methods?

Reference:

[1] Ajay, A., Du, Y., Gupta, A., Tenenbaum, J., Jaakkola, T., & Agrawal, P. (2022). Is conditional generative modeling all you need for decision-making? arXiv preprint arXiv:2211.15657.

[2] Liang, Z., Mu, Y., Ding, M., Ni, F., Tomizuka, M., & Luo, P. (2023). Adaptdiffuser: Diffusion models as adaptive self-evolving planners. arXiv preprint arXiv:2302.01877.

**Limitations:**

Yes, the authors discussed the limitations in their paper.

---

> ### Author Rebuttal · Authors · 2024-08-07
>
> Thank you for your valuable feedback, which can enhance our manuscript.
>
> > **(Weakness 1)** Lack of theoretical guarantee to support the algorithm: The authors only provide empirical results, lacking theoretical analysis of the performance of the proposed method.
>
> While existing work [1] proposes theoretical bounds between conditioning value and generated value, these proofs rely on stringent assumptions, which may not align with complex tasks in Offline RL benchmarks. Empirically, on GTA, we conducted an analysis of the correlation between the conditioning value and the rewards of generated trajectories. As illustrated in Figure 1, attached in pdf, we observed Pearson correlation coefficients of 0.55, 0.91, and 0.99 for halfcheetah-medium, medium-replay, and medium-expert datasets, suggesting that GTA can generate trajectories aligned with the conditioned return.  Augmented datasets with high-rewarding trajectories can aid offline RL policies in reaching higher performance.
>
> GTA has demonstrated consistent performance gains through extensive experiments across various environments. Thorough ablation studies have further proven the effectiveness of our proposed method. We would like to emphasize the empirical results presented in our paper, which underscore the effectiveness of our approach.
>
>
> [1] Yuan, Hui, et al. "Reward-directed conditional diffusion: Provable distribution estimation and reward improvement." Advances in Neural Information Processing Systems 36 (2024).
>
>
> > **(Question 1)** How to select the hyperparameter $\alpha$ in the denoising with amplified return guidance in equation 8? **(Question 3-1)** Is the proposed method sensitive to the hyperparameter $\mu$?
>
> As discussed in the general response, GTA outperforms other baselines in locomotion tasks even with a single hyperparameter configuration ($\mu=0.25, \alpha=1.1$), demonstrating the robustness of our method. We also provide a general recipe for selecting hyperparameters for new tasks: Increasing $\mu$ and $\alpha$ for datasets with low quality to promote further performance gain.
>
> > **(Question 2)** In the generation phase, do you generate whole trajectories or generate subsequences of trajectories?
>
> Instead of generating the entire trajectory, we focus on generating subtrajectories. Please refer to Table 8 in Appendix E.1 for more details about subtrajectory generation.
>
> > **(Question 3-2)** Based on the description in Figure 4 and Appendix F.3, when \mu = 1, it only performs well on the dataset with low quality. Does GTA have trouble preserving and reconstructing the high-rewarding trajectories?
>
>
>
> In addition to the dynamic MSE and oracle reward presented in Appendix F.4, we include data quality metrics for trajectories generated with a higher $\mu$ in the table below. The results indicate that while a $\mu$ of 0.75 results in a higher dynamic MSE than a $\mu$ of 0.25, it still maintains a lower value than SynthER. This demonstrates that GTA does not encounter significant issues in reconstructing high-rewarding trajectories.
>
> Table 1. Dynamic MSE of generated data on halfcheetah-medium-expert-v2.
> ||SynthER|GTA($\mu=0.25, \alpha=1.3$)|GTA($\mu=0.75, \alpha=1.3$)|
> |-|-|-|-|
> |Dynamic MSE($\times 1e-2$)|0.91|0.79|0.90|
> |Optimality|7.7|10.07|11.56|
>
>
>
>
> > **(Question 4)** Can you provide comparison results with diffusion-based offline RL methods? For example, DecisionDiffuser and AdaptDiffuser.
>
> We provide a comparison between diffusion planners [2, 3, 4] and GTA in the table 1, 2 attached in PDF. As shown in the table, GTA outperforms diffusion planners in locomotion tasks and demonstrates significant efficiency in test-time evaluation.
>
> As mentioned in related work section, our method resembles diffusion planners in terms of generating trajectories with high return guidance. However, adopting diffusion as an augmentation method and utilize the model-free offline RL policies can transfer extensive computational burden of diffusion models from the evaluation stage to the data preparation stage. It can bypass the crucial challenge of diffusion planners, which requires extensive cost for sampling action.
>
> [2] Janner, Michael, et al. "Planning with diffusion for flexible behavior synthesis." arXiv preprint arXiv:2205.09991 (2022).
>
> [3] Ajay, Anurag, et al. "Is conditional generative modeling all you need for decision-making?." arXiv preprint arXiv:2211.15657 (2022).
>
> [4] Liang, Zhixuan, et al. "Adaptdiffuser: Diffusion models as adaptive self-evolving planners." arXiv preprint arXiv:2302.01877 (2023).
>
>
> > **(Question 5)** What is the baseline algorithm in the ablation study of reweighted sampling?
>
> We choose TD3BC for the ablation study of reweighted sampling because TD3BC requires minimal computational cost along the offline RL baselines (IQL, CQL, MCQ).
>
> >**(Question 6)** Can you also apply the reweighted sampling techniques for other baseline methods?
>
> We conduct experiments on the effect of reweighted sampling for other baselines. Specifically, we sample transition from the buffer with probability proportional to its reward during policy training. Table 1 describes the performance of reweighted sampling on other baselines. We find that baselines with reweighted sampling underperform GTA.
>
> Table 1. Experiment results of reweighted sampling on baselines.
> ||Original|S4RL|SynthER|GTA|
> |-|-|-|-|-|
> |Halfcheetah-medium|48.52±0.39|48.70±0.31|48.73±0.83|57.84±0.51|

---

> > ### Comment · Area_Chair_stFo · 2024-08-11
> >
> > Dear Reviewer UiuM, thanks for giving the authors a detailed list of questions. Did the authors answer these to your satisfaction or did they raise any concerns for you about accepting this paper? If so, it would be great to clarify while we can still interact with the authors. Thanks!

---

> > ### Comment · Reviewer_UiuM · 2024-08-12
> >
> > I appreciate the detailed response, most of my questions have been addressed. Regarding the ablation study against \alpha and \mu, and the comparison results with reweighted sampling, I believe they should be evaluated more comprehensively with more testing values and tasks. Considering the limited time of the rebuttal phase, I understand that the authors can not provide the full results. However, I suggest to have a more detailed discussion regarding this part in their revision.
> >
> > After reading the reviews from other reviewers and the authors' responses, I decided to raise my score from 5 to 6 in favor of acceptance.

---

> > > ### Author Response · Authors · 2024-08-13
> > >
> > > Thank you for your thoughtful response and detailed review of our rebuttal. Following your suggestion, we will expand the ablation study presented in Appendix G by conducting additional experiments across various tasks and incorporating these results into the revised manuscript.
> > >
> > > If you have any further suggestions or considerations, please feel free to share them with us. Once again, we sincerely appreciate your insightful feedback.

---

### Official Review · Reviewer_hVee · 2024-07-11

**Soundness:** 3
**Presentation:** 3
**Contribution:** 3
**Rating:** 6
**Confidence:** 4

**Summary:**

To improve the quality of offline datasets, in this paper, a generative data augmentation approach is proposed by leveraging the diffusion models. Moreover, with the adoption of partial noising and denoising framework with amplified return guidance, the trajectory can be guided towards high-rewarding region. Finally, with the generated trajectories, existing offline RL methods can be utilized to learn the optimal policy.

**Strengths:**

The paper is clearly organized and the experimental results show the effectiveness of the proposed approach. Also, the separation of the data generation with diffusion models and the policy learning mitigates the possible time and computation cost in the policy learning. Moreover, the trajectory-level data generation can capture the transition dynamics, which is beneficial for environments with sparse rewards.

**Weaknesses:**

The proposed approach is mainly a direct combination of current techniques, e.g., the diffusion model for data generation, the adding noise for the exploration and the existing offline RL techniques for policy learning, the theoretical contribution of the paper is trivial. Also, experimental results are still not sufficient, e.g., the lack of the comparations of time and computation costs between different methods, and only two data augmentation baselines are compared. Moreover, the proposed approach is limited to trajectories with reward signals, which is not the case in many real-world applications. Furthermore, it is unclear to me whether there are any theoretical guarantees to push the generated trajectory toward the high-rewarding region.

**Questions:**

1. In section 4.2, by denoising with amplified return guidance, the authors claim the trajectory can be guided towards high-rewarding regions, are there any theoretical guarantee for this statement?
2. How about the whole computation and time cost of the proposed approach when it is compared with other diffusion-free baselines?
3. As shown in the experiments, for different environments, different noise levels are required for the optimal performance, then how can we determine the noise level for a new task?
4. The proposed approach aims to deal with trajectories with reward signals, however, in reality, it may be difficult to obtain such signals, can the proposed approaches be extended to the cases where no reward signals can be provided?

**Limitations:**

The authors have addressed some limitations, but for some other limitations, e.g., the computation and time cost of the proposed approach, the application of the proposed approach in some real-world applications with no reward signal provided should be further stated.

---

> ### Author Rebuttal · Authors · 2024-08-07
>
> Thank you for your critical reviews and valuable feedback.
>
> > **(Weakness 1)** The proposed approach is mainly a direct combination of current techniques.
>
> Please note that we propose the hypothesis that augmenting the offline dataset with feasible and high-rewarding trajectories would boost the performance of offline RL algorithms. To achieve this, we made a novel combination of various techniques. GTA involves (1) generating trajectory-level data to capture sequential relationships between transitions and long-term transition dynamics, (2) introducing partial noising to control the exploration level of generated trajectories, and (3) guiding diffusion model with amplified return to find high-rewarding trajectories. We effectively balance exploration and exploitation with two strategies carefully tailored for diffusion models, as both strategies can seamlessly be integrated with the forward and reverse process of diffusion models. Our method consistently outperforms several baselines across different environments, highlighting the effectiveness of our proposed method. We want to emphasize that numerous machine learning studies continue to explore innovative combinations of established techniques.
>
> > **(Question 1)** In section 4.2, by denoising with amplified return guidance, the authors claim the trajectory can be guided towards high-rewarding regions, are there any theoretical guarantee for this statement?
>
> There is existing work [1] proposing theoretical bounds between conditioning value and generated value. However, these proofs rely on stringent assumptions, making it challenging to directly apply them to GTA. Meanwhile, related works with return guidance [2, 3, 4] show that the conditionally sampled trajectories yield high rewards, demonstrating the effectiveness of return-guidance.
>
> To verify that our conditional sampling indeed generates high-reward trajectories, we conducted analysis on the correlation between the conditioning value and the rewards of generated trajectories. As illustrated in Figure 1 attached in pdf, we observed Pearson correlation coefficients of 0.55, 0.91, and 0.99 for halfcheetah-medium, medium-replay, and medium-expert dataset, respectively. This indicates that diffusion model effectively samples trajectories aligned with the conditioned return.
>
> [1] Reward-directed conditional diffusion: provable distribution estimation and reward improvement
>
> [2] Janner, Michael, et al. "Planning with diffusion for flexible behavior synthesis." arXiv preprint arXiv:2205.09991 (2022).
>
> [3] Ajay, Anurag, et al. "Is Conditional Generative Modeling all you need for Decision Making?." The Eleventh International Conference on Learning Representations.
>
> [4] Liang, Zhixuan, et al. "AdaptDiffuser: Diffusion Models as Adaptive Self-evolving Planners." International Conference on Machine Learning. PMLR, 2023.
>
> > **(Question 2)** How about the whole computation and time cost of the proposed approach when it is compared with other diffusion-free baselines?
>
> As mentioned in general response, we compare the whole computation and time cost of the proposed approach with other diffusion-free baselines. For model-based methods, we need to train dynamic model and generate synthetic trajectories during policy training.
>
> The table 2 attached in PDF summarizes the whole training and evaluation time cost of algorithms. While training dynamic models takes shorter time than training diffusion models, model-based RL takes much longer time for training policy as it rollouts synthetic trajectories during training.
>
>
> > **(Question 3)** As shown in the experiments, for different environments, different noise levels are required for the optimal performance, then how can we determine the noise level for a new task?
>
> As mentioned in the general response, we find that even if we fix $\mu$ across different environments, we achieve higher performance than other baselines. We also observe that we can improve the performance further if we increase $\mu$ for exploration when we have low-quality dataset. Therefore, we recommend selecting $\mu$ based on dataset quality for a new task instead of extensive online tuning.
>
> > **(Question 4)** The proposed approach aims to deal with trajectories with reward signals, however, in reality, it may be difficult to obtain such signals, can the proposed approaches be extended to the cases where no reward signals can be provided?
>
> Numerous offline RL methods [5, 6, 7, 8, 9] aim to learn decision-making policy that maximizes expected cumulative rewards. In this context, we want to emphasize that GTA focuses on enhancing the optimality of offline datasets while minimizing the degradation of dynamic plausibility to improve the performance of offline RL algorithms. Therefore, generating trajectories without any reward signal is beyond our scope, as they cannot be augmented toward high-rewarding trajectories and are not the objective of RL problems. Please acknowledge that we conduct extensive experiments on realistic and challenging tasks such as AntMaze, FrankaKitchen, Adroit, and pixel-based environments.
>
> [5] Fujimoto, Scott, and Shixiang Shane Gu. "A minimalist approach to offline reinforcement learning." Advances in neural information processing systems 34 (2021): 20132-20145.
>
> [6] Kumar, Aviral, et al. "Conservative q-learning for offline reinforcement learning." Advances in Neural Information Processing Systems 33 (2020): 1179-1191.
>
> [7] Kostrikov, Ilya, Ashvin Nair, and Sergey Levine. "Offline Reinforcement Learning with Implicit Q-Learning." International Conference on Learning Representations.
>
> [8] Lyu, Jiafei, et al. "Mildly conservative q-learning for offline reinforcement learning." Advances in Neural Information Processing Systems 35 (2022): 1711-1724.
>
> [9] Yu, Tianhe, et al. "Combo: Conservative offline model-based policy optimization." Advances in neural information processing systems 34 (2021): 28954-28967.

---

> > ### Comment · Reviewer_hVee · 2024-08-11
> >
> > I appreciate the authors' detailed response and additional experimental results，most of my concerns have been addressed, I'd like to raise my score to 6, best.

---

> > > ### Author Response · Authors · 2024-08-11
> > >
> > > Thank you for your kind response and for thoroughly reviewing our rebuttal. While most of your concerns have been addressed, if there are any remaining issues or points of discussion, please feel free to share them with us. We are always ready to engage in further discussion. Once again, we appreciate your thoughtful feedback.

---

### Official Review · Reviewer_ABdH · 2024-07-11

**Soundness:** 3
**Presentation:** 3
**Contribution:** 3
**Rating:** 7
**Confidence:** 4

**Summary:**

This paper builds on ideas from SynthER but adds classifier free guidance to boost the returns of the generated trajectories. This makes sense as the resulting data has higher quality and the paper is a totally sensible next step in the series of works building upon SynthER, which is an exciting direction for research. The paper is well executed and a solid contribution.

**Strengths:**

* A highly relevant area building on synthetic data generation for offline RL. Out of all the current active areas in RL research this one benefits the most from the current foundation model/large data regime, and scaling offline RL has been shown to be highly impactful in areas such as Robotics (E.g. RTX).
* The method is not overly complicated and builds upon the recent SynthER paper, presented at NeurIPS 2023. This is an example of a simple idea that makes a great deal of sense, and it is well executed.
* The paper reads well.
* The experiments go beyond what was done in SynthER, including some new environments. It is great to see this - as RL needs to keep pushing for more complex benchmarks and not just sticking to D4RL and Atari "because thats what the previous paper did".

**Weaknesses:**

There are no major weaknesses here, it is a solid paper taking a nice step in an exciting general direction of research. The below points are fairly minor:
* The authors could cite Ball et al 2021 "Augmented World Models" as another example of data augmentation in offline RL.
* The authors could discuss how this relates to another paper building on SynthER, Policy-Guided Diffusion by Jackson et al. This is very recent work so makes sense it is not needed as a baseline, but the comparison may be worth including in text.

**Questions:**

NA

**Limitations:**

Nice to see this in the main body rather than the Appendix. It would be good to add a discussion on scalability if possible.

---

> ### Author Rebuttal · Authors · 2024-08-07
>
> Thank you for your positive review and valuable feedback!
>
> > **(Weakness 1)** The authors could cite Ball et al 2021 "Augmented World Models" as another example of data augmentation in offline RL.
>
> Thank you for pointing out crucial paper. The paper [1] augments learned model with simple transformations and approximate the augmentation for unseen environment in a self-supervised fasion for zero-shot generalization. While we focus on a single task offline RL problems, it is highly related in terms of data augmentation methods for offline RL problems. We will cite the paper in the related work section.
>
> > **(Weakness 2)** The authors could discuss how this relates to another paper building on SynthER, Policy-Guided Diffusion by Jackson et al. This is very recent work so makes sense it is not needed as a baseline, but the comparison may be worth including in text.
>
> Thank you for highlighting an exciting concurrent work. Policy-Guided Diffusion (PGD) [2] generates synthetic trajectories for augmentation with classifier guidance from the target policy. On the other hand, GTA generates trajectories from the original trajectory, and introduce partial noising and denoising framework. To generate high-rewarding trajectories, GTA uses amplified return guidance. While the method is slightly different, both papers share ultimate objective, generating high-rewarding trajectories while retaining low dynamics error to improve offline RL algorithms. We will add comparison with PGD in the manuscript.
>
> [1] Ball, Philip J., et al. "Augmented world models facilitate zero-shot dynamics generalization from a single offline environment." International Conference on Machine Learning. PMLR, 2021.
>
> [2] Jackson, Matthew Thomas, et al. "Policy-guided diffusion." arXiv preprint arXiv:2404.06356 (2024).

---

> > ### Comment · Reviewer_ABdH · 2024-08-10
> > **SGTM**
> >
> > Thanks for the response!

---

> > > ### Author Response · Authors · 2024-08-11
> > >
> > > Thank you for your kind response! Please let me know if you have any further suggestions or adjustments you would like us to consider.

---

### Official Review · Reviewer_WvGz · 2024-07-12

**Soundness:** 3
**Presentation:** 3
**Contribution:** 3
**Rating:** 6
**Confidence:** 4

**Summary:**

The paper introduces Generative Trajectory Augmentation (GTA), a data augmentation approach for Offline Reinforcement Learning (RL) that enhances the quality of static datasets by generating high-rewarding and dynamically plausible trajectories using a conditional diffusion model. GTA partially noises original trajectories and then denoises them with classifier-free guidance via conditioning on amplified return value. The authors demonstrate that GTA improves the performance of various offline RL algorithms across several benchmark tasks.

**Strengths:**

- GTA integrates seamlessly with existing offline RL methods. It builds on previous advancements, and new offline RL algorithms could also benefit from it, given its agnostic nature towards the specific RL method used.
- The ability to generate high-return trajectories that do not exist in the logged data is a significant advantage. This feature potentially improves the performance of offline RL algorithms by enriching the dataset with valuable transitions, as supported by the experimental results.
- The experimental results presented in the paper show significant improvements across various offline RL algorithms and environments. Additionally, thorough ablations are provided, highlighting the impact of different components and hyperparameters on the overall performance. This comprehensive evaluation demonstrates the practical effectiveness of GTA.

**Weaknesses:**

- The claim that GTA-generated data adheres to the dynamics of the environment (lines 145-148) seems unfounded. Is there a principled argument on why diffusion models would learn the dynamics of the environment well? Especially where the goal is to create transitions outside the dataset. Figure 5 does not conclusively show that GTA is dynamically plausible, especially for tasks like Cheetah compared to methods like S4RL.
- Compared to S4RL, the augmented trajectories appear less plausible. This issue is evident in tasks like HalfCheetah, yet GTA still achieves better rewards. It is unclear why this does not affect the final results.
- Based on Appendix B.2, the performance of GTA appears to depend heavily on finely-tuned hyperparameters, as suggested by the different values used for different environments in Table 6. This raises concerns about the generalizability of the method, indicating that it might rely on online manual tuning.

**Questions:**

- How do you ensure that the generated trajectories are plausible and adhere to the environment's dynamics?  What mechanisms to pick &alpha to handle cases where the return used for high-reward guidance is not reasonable? How do you ensure that the generated trajectories remain valid and useful?
- Does each task require a new &mu, &alpha hyperparameter? How does this impact the generalizability and practical application of GTA?
- What is the proportion of augmented trajectories to original ones in the final dataset? Is the entire offline dataset kept before training, with augmented trajectories added on top? Do all the offline datasets have the same size? Are 5 million augmented tranitions added regardless of the original dataset's size?
- Could you clarify if Figure 3 is based on a real example or if it is just an illustration of the method's intuition? If it is the latter, how can you ensure it accurately represents the method's practical application?

**Limitations:**

The authors discuss the limitations and potential impacts in their paper.

---

> ### Author Rebuttal · Authors · 2024-08-07
>
> Thank you for your positive review and valuable feedback!
> > **(Weakness 1-1)**  Is there a principled argument on why diffusion models would learn the dynamics of the environment well? Especially where the goal is to create transitions outside the dataset. **(Question 1-1)** How do you ensure that the generated trajectories are plausible and adhere to the environment's dynamics?
>
> As you correctly pointed out, diffusion model of GTA does not explicitly ensure dynamic plausibility. Instead, it implicitly learns the dynamics governing the trajectory through learning the data distribution.
>
> GTA concentrates on enhancing the optimality of the dataset while reducing the degradation of the environment's dynamics of the augmented dataset to improve the performance of offline RL algorithms. Therefore, we will adjust our statement from "ensuring dynamic plausibility" to "minimizing the degradation of dynamic plausibility". We apologize for the overclaiming of our proposed method.
>
> However, our method still has the potential to generate high-rewarding and dynamically plausible trajectories by virtue of two novel strategies: *partial noising and denoising framework* and *amplified return guidance*. As discussed in Appendix F.1, amplified return conditioning prevents the partially noised trajectories denoise conditioning on excessively high value, yielding lower dynamic MSE. Additionally, table 1 shows the dynamic mse of the generated dataset according to the differing level of $\mu$. The result demonstrates that low $\mu$ does help to lose dynamic information less.
>
> Table 1. Dynamic MSE of generated data on walker-medium-v2.
> |$\mu$|Dynamic MSE($\times 1e-2$)|
> |-|-|
> |0.1|2.80|
> |0.25|3.02|
> |0.5|3.04|
> |0.75|3.34|
> |1.0|3.32|
>
> To sum up, even though GTA does not explicitly enforce the model to generate dynamically plausible dataset, we introduce *partial noising and denoising framework* and *amplified return guidance* to minimize the degradation of dynamics of generated data.
>
> > **(Weakness 1-2, Weakness 2)** Figure 5 does not conclusively show that GTA is dynamically plausible, especially for tasks like Cheetah compared to methods like S4RL, yet GTA still achieves better rewards. It is unclear why this does not affect the final result.
>
> As the reviewer noted, the trajectories generated by GTA exhibit larger dynamic MSE than S4RL in some cases like Halfcheetah. However, we observe that the dynamic plausibility of GTA is still better than other baselines such as SynthER. More importantly, GTA augments trajectories with higher rewards, which significantly affects the performance of offline RL policies.
>
> > **(Weakness 3)** The performance of GTA appears to depend heavily on finely-tuned hyperparameters. **(Question 2)** Does each task require a new &mu, &alpha?
>
> To address the concern that the reviewer mentioned, we compared GTA with other baselines on single set parameter($\alpha=1.1, \mu=0.25$). The results demonstrate that fixed parameter configurations generally show performance gain across locomotion tasks, indicating that it may not require extensive online tuning. Furthermore, we propose a guideline for determining the appropriate $\mu$ and $\alpha$ for new tasks to boost performance. Please refer to the general response for a more detailed description.
>
> > **(Question 1-2)** What mechanisms to pick &alpha to handle cases where the return used for high-reward guidance is not reasonable? How do you ensure that the generated trajectories remain valid and useful?
>
> As you mentioned, excessively guiding towards high rewards can result in invalid trajectories. This phenomenon occurs, as shown in Table 17 in Appendix F.1, when we generate trajectories conditioned on the maximum return that can be achieved from the environment. To prevent conditioning on unreasonable return, we introduced amplified return guidance, which involves multiplying the original trajectory by $\alpha$ for conditioning. It controls the exploitation level of GTA, and we observe that our method empirically generates valid and useful trajectories to improve the performance of RL algorithms.
>
> > **(Question 3)** What is the proportion of augmented trajectories to original ones? Is the entire offline dataset added augmented trajectories on top? Do all the offline datasets have the same size? Are 5M augmented transitions added regardless of the original dataset's size?
>
> GTA augments 5M transitions for D4RL, and 1M transitions for VD4RL regardless of different sizes of original datasets. We follow the procedure of the prior method SynthER for a fair comparison. After the augmentation, we added augmented trajectories to the original dataset. There is no reason to exclude the original dataset, as it preserves the true dynamics of the environment. Please refer to Table 10 of Appendix G.4 for more discussion on the size of the augmented dataset.
>
> > **(Question 4)** Could you clarify if Figure 3 is based on a real example or if it is just an illustration of the method's intuition? If it is the latter, how can you ensure it accurately represents the method's practical application?
>
> While Figure 3 of section 4.2 is presented for intuitive comprehension, our empirical results of table 2 clearly demonstrate the relationship between $\mu$, oracle reward, and the deviation from the original trajectory. Table 2 shows that both oracle reward and deviation from the original dataset tend to increase as $\mu$ becomes larger. These results align well with Figure 3, which illustrates that trajectories deviate from the original trajectories and towards high-rewarding regions as $\mu$ gets larger.
>
> Table 2. Analysis of dataset generated by GTA with different $\mu$.
> |$\alpha$ = 1.4|Oracle Reward|Deviations($\times 1e-1$)|
> |-|-|-|
> |offline data|4.77|0.00|
> |$\mu$ = 0.1|4.84|0.01|
> |$\mu$ = 0.25|5.06|0.24|
> |$\mu$ = 0.5|7.21|11.31|
> |$\mu$ = 0.75|7.23|21.50|
> |$\mu$ = 1.0|7.24|21.99|

---

> > ### Comment · Reviewer_WvGz · 2024-08-11
> >
> > Thank you for your answers.

---

> > > ### Author Response · Authors · 2024-08-11
> > >
> > > We would like to express our gratitude once again for your feedback, which led our paper to a more progressive and positive direction. In response to your comments, we will revise our paper to include additional details about Figure 5 and provide practitioner guidance for the hyperparameter settings.
> > >
> > > We are fully prepared to respond to any additional discussions or inquiries you may have, so please feel free to reach out at any time. Thank you.

---

### Official Review · Reviewer_cNng · 2024-07-27

**Soundness:** 3
**Presentation:** 3
**Contribution:** 2
**Rating:** 6
**Confidence:** 4

**Summary:**

The paper presents Generative Trajectory Augmentation (GTA) that is aimed at improving offline reinforcement learning (RL). GTA uses a diffusion model conditioned on high returns to generate high-rewarding trajectories. These trajectories are used to augment the static datasets used to train offline RL algorithms. The augmentation process involves partially adding noise to existing trajectories and then refining them using a denoising mechanism guided by an amplified return value. The authors demonstrate that GTA improves the performance of popular offline RL algorithms across tasks in the D4RL benchmark. They also analyze the quality of the augmented trajectories, showing improvements in both data optimality and novelty over two baselines (S4RL and SynthER).

**Strengths:**

**Originality:** The use of a conditional diffusion model for trajectory augmentation is a novel approach that adds computation overhead on the data creation stage instead of the policy learning stage.

**Quality:** The paper includes experiments across tasks from the D4RL benchmark, demonstrating the effectiveness of GTA in various settings, including dense and sparse reward tasks, high-dimensional robotics tasks, and a pixel-based observation task.

**Clarity:** The paper is well-organized and presents the methodology, experiments, and results clearly and logically. The authors provide an anonymized link to their code enhancing the reproducibility and transparency of the proposed method.

**Significance:** GTA is shown to be compatible with offline RL algorithms, making it a flexible solution for data augmentation in offline RL. By adding novel high-rewarding trajectories to the datasets in the D4RL benchmark, GTA shows improvement in the performance of offline RL algorithms.

**Weaknesses:**

1. As shown in Figures 4 (a)(b) and Table 25, the performance of GTA is sensitive to the choice of hyperparameters, such as the noising ratio ($\mu$) and the multiplier for conditioned return ($\alpha$) which might require extensive tuning for different tasks.
2. While empirical results show that the generated trajectories using GTA are dynamically plausible (Figure 5), it is not explicitly enforced in GTA and might be a byproduct of generating high-rewarding trajectories. The reviewer believes that further evaluation of GTA on real-world data is needed to make claims regarding GTA's dynamic plausibility such as in lines 6-9 (“In response, we introduce … and dynamically plausible”), lines 39-40 (“GTA is designed … dynamic plausibility”).
3. Related to weakness #2, while the paper shows GTA's effectiveness on standard benchmarks, it lacks validation in real-world applications where the dynamics and data distributions may differ significantly from simulated environments. This poses a question on the real-world applicability of GTA.
4. The paper does not include any experiments comparing GTA with model-based RL baselines. Although the authors state that both approaches perform data augmentation at different stages (data generation vs policy learning), the reviewer believes that a comparative study should be included. This is because both approaches learn a separate model to generate augmented data.

Minor:
- Line 48: “... any offline RL algorithms …” should be “... any offline RL algorithm …”.
- A pink curve is present in Figure 4 (b), but its legend is missing. The blue curve denoting $\mu$ = 1.0) is missing in this figure,

**Questions:**

1. Why is IQL the only chosen baseline (and not TD3-BC, CQL, MCQ) for the tasks listed in Table 2 (Adroit and FrankaKitchen)?
2. Why is DrQ+BC the only chosen baseline and Cheetah-run the only chosen pixel-based observation task in Table 3? Why were results on the other baselines (IQL, TD3-BC, CQL, and MCQ) not presented for the pixel-based task(s)?
3. How compute-intensive (GPU or hours) is running GTA on the pixel-based observation tasks?

**Limitations:**

1. As the authors have mentioned, while empirical results on the D4RL benchmark tasks show that GTA has low dynamic MSE, in tasks where dynamic violations have a critical impact, the performance boost may not be significant.
2. The current evaluation is limited to simulated tasks, and the effectiveness of GTA in real-world offline RL tasks remains to be validated.
3. As shown in Figures 4 (a)(b) and Table 25, the performance of GTA is sensitive to the hyperparameters ($\mu$ and $\alpha$) which might limit its applicability.

---

> ### Author Rebuttal · Authors · 2024-08-07
>
> Thank you for the insightful review!
>
> > **(Weakness 1, Limitation 3)** The performance of GTA is sensitive to the choice of hyperparameters, which might require extensive tuning for different tasks.
>
> As we illustrated in tables in general response, we demonstrate that GTA outperforms other data augmentation baselines with a single set of hyperparameters across locomotion tasks. This result indicates that we may not need to extensively tune such hyperparameters. Furthermore, we also provide a general recipe for selecting hyperparameters for a new task: choosing high $\mu$ and $\alpha$ for low-quality datasets to promote exploration.
>
> > **(Weakness 2-1)** Dynamic plausibility is not explicitly enforced in GTA and might be a byproduct of generating high-rewarding trajectories.
>
> As you mentioned, GTA does not explicitly enforce dynamic plausibility. Instead, it implicitly learns the dynamics through the trajectories which inherit true dynamics of environment, resulting in the generation of plausible trajectories.
>
> We want to clarify that dynamic plausibility is not a byproduct of generating high-rewarding trajectories. Table 1 compares the dynamic MSE of total augmented trajectories to that of high-rewarding trajectories (top 10% in terms of reward). We found that the dynamic MSE of both cases are similar, validating our claim.
>
> Table 1. Dynamic MSE of data generated with GTA using halfcheetah-medium-v2
> ||total trajectories|high rewarding trajectories (top 10%)|
> |-|-|-|
> |Dynamic MSE($\times 1e-2$)|0.843|0.846|
>
>
>
> > **(Weakness 2-2, 3, Limitation 2)** GTA lacks validation in real-world applications where the dynamics and data distributions may differ significantly.
>
>
> Thank you for suggesting problem setting that enhaces our manuscript . To verify the effectivenes of GTA in real-world data, we conduct further experiments on the NeoRL benchmark [1]. NeoRL is composed of datasets with narrower data distributions, enforced stochasticity, and aleatoric uncertainty to evaluate how offline RL algorithms behave in realistic environments.
>
> We train GTA on the halfcheetah-v3-medium-noise-1000 and halfcheetah-v3-low-noise-1000 of NeoRL benchmark to evaluate the performance. As shown in table 2, GTA enhances the performance of the base algorithm, indicating that GTA can also generate trajectories with real-world datasets.
>
> Table 2. Experiment results on NeoRL benchmark. Experiments are conducted with 4 random seeds. Base algorithm is CQL.
> | Env | Original | S4RL | GTA |
> |:- |:- |:- |:- |
> | Halfcheetah-v3-L-1000 | 4352.58 ± 79.81| 4320.42 ± 76.65| **4359.08 ± 46.74**|
> | Halfcheetah-v3-M-1000 | 6162.49 ± 713.11 | 5759.13 ± 224.87 | **6173.46 ± 739.10** |
>
> [1] Qin, Rong-Jun, et al. "NeoRL: A near real-world benchmark for offline reinforcement learning." Advances in Neural Information Processing Systems 35 (2022): 24753-24765.
>
>
> > **(Weakness 4)** The paper does not include any experiments comparing GTA with model-based RL baselines.
>
> Thank you for your insightful comment. As illustrated in the general response, we compare offline model-based RL methods, such as MOPO, MOReL, and COMBO, with GTA. We take the results of baselines from their original papers. For GTA, we report the score of TD3BC and CQL as base offline RL algorithms.
>
> As shown in Table 1 of the attached PDF, GTA outperforms offline model-based RL algorithms in locomotion tasks. It suggests that using a diffusion model to augment trajectories is more beneficial than learning and exploiting a single-transition dynamic model [2, 3].
>
> [2] Janner, Michael, et al. "Planning with Diffusion for Flexible Behavior Synthesis." International Conference on Machine Learning. PMLR, 2022.
>
> [3] Jackson, Matthew Thomas, et al. "Policy-guided diffusion." arXiv preprint arXiv:2404.06356 (2024).
>
> > **(Question 1)** Why is IQL the only chosen baseline for the Adroit and FrankaKitchen?
>
> For Adroit and FrankaKitchen tasks, IQL is the most reliable baseline among offline RL algorithms we adopted (TD3BC, MCQ, CQL). TD3BC exhibits near-zero performance on these tasks. MCQ does not provide any hyperparameter setting for these tasks, while the main hyperparameter $\tau$ in MCQ significantly influences the performance of the algorithm. Although CQL offers hyperparameter settings for these tasks, its training is notoriously unstable, leading to significant performance fluctuations.
>
> > **(Question 2)** Why is DrQ+BC the only chosen baseline and Cheetah-run the only chosen pixel-based observation task in Table 3? Why were results on the other baselines (IQL, TD3-BC, CQL, and MCQ) not presented for the pixel-based task(s)?
>
> To evaluate GTA on more diverse algorithms and environments, we additionally conduct experiments on walker-walk environment and add BC algorithm for baseline as done in [4]. As shown in table 3, we achieve better performance than other baselines across both environments and algorithms, indicating the generalizability of our method.
>
> Table 3. Experiment results on pixel-based observation tasks. The mean scores of table are calculated with 3 seeds.
> ||BC|||DrQ+BC|||
> |:-|:-|-|-|-|-|-|
> || Original | SynthER | GTA| Original | SynthER | GTA |
> | Cheetah-run | 40.0 | 30.0| **41.1** | 45.8 | 31.2| **49.8**|
> | Walker-walk | 37.8 | 24.4| **40.0** | 39.7 | 23.8| **41.1**|
>
> [4] Lu, Cong, et al. "Synthetic experience replay." Advances in Neural Information Processing Systems 36 (2024).
>
>
> > **(Question 3)** How compute-intensive (GPU or hours) is running GTA on the pixel-based observation tasks?
>
> Please note that the diffusion model for pixel-based observation tasks generates a latent vector rather than raw pixel observation. In a single RTX 3090, training GTA in a pixel-based environment for 500k gradient steps requires 50 hours.
>
> We promise to fix minor issues raised by the reviewer of our manuscript.

---

> ### Comment · Reviewer_cNng · 2024-08-13
>
> Thank you for your detailed response and for providing additional supporting and clarifying empirical results. In line with reviewer **WvGz**, I am still skeptical about the paper's claims regarding GTA ensuring "dynamic plausibility". I would be more comfortable if such a claim is toned down; for instance, augmented trajectories generated from benchmark datasets using GTA exhibit dynamic plausibility.
>
> I am satisfied with the author's responses to the rest of my questions and concerns and have raised my rating from 5 to 6.

---

> > ### Author Response · Authors · 2024-08-13
> >
> > We would like to express our gratitude for your thorough review and thoughtful comments on our rebuttal.
> >
> > Following your suggestion, we will tone down our statement from "ensuring dynamic plausibility" to "minimizing the degradation of dynamic plausibility". We apologize for the overclaiming of our proposed method.
> >
> > If you have any additional suggestions or considerations, please feel free to share them with us. Once again, we sincerely appreciate your thoughtful feedback.

---

### Author Rebuttal · Authors · 2024-08-07

We sincerely thank the review committee for their detailed feedback. We appreciate the recognition of our paper's strengths, highlighted by the reviewers: **Originality** (CNng, ABdH), **Significance** (CNng, WvGZ, ABdH, UiuM), and **Extensive experiments** (CNng, WvGz, ABdH, UiuM). In response to the reviewers' feedback, we provide a summary of conducted additional experiments:

- **Comparison with offline model-based RL and diffusion planners (Table 1)**

We present a table that compares the performance of GTA with model-based RL and diffusion planners in locomotion tasks. We observe that GTA outperforms recent baselines in terms of average performance.

- **Analysis on computational cost (Table 2)**

We present a table that shows total computational time for training GTA and other methods. We find that while training cost of GTA is relatively high, the evaluation time is much faster than other methods, especially when compared with diffusion planners.

- **Guidance for selecting $\mu$ and $\alpha$ for a new task**

We would like to address the concerns regarding the generalizability of GTA to new tasks, specifically on hyperparameter settings for noise level $\mu$ and guidance multiplier $\alpha$.


We conducted experiments across gym locomotion environments using a single set of hyperparameters ($\mu=0.25, \alpha=1.1$). As shown in the table, GTA outperforms SynthER even with a single hyperparameter setting, highlighting the generalizability of our method.


Table 1. D4RL normalized score on locomotion environments with fixed $\alpha$ and $\mu$. The experiments are conducted with TD3BC.

| Env             | None          | SynthER       | GTA($\mu = 0.25, \alpha=1.1$) |
| --------------- | ------------- | ------------- | ----------------------------- |
| halfcheetah-m-r | 44.64 ± 0.71  | 45.57 ± 0.34  | **46.48 ± 0.39**              |
| halfcheetah-m   | 48.42 ± 0.62  | 49.16 ± 0.39  | **49.22 ± 0.52**              |
| halfcheetah-m-e | 89.48 ± 5.50  | 85.47 ± 11.35 | **94.98 ± 1.66**              |
| hopper-m-r      | 65.69 ± 24.41 | **78.81 ± 15.80** | 72.86 ± 28.42             |
| hopper-m        | 61.04 ± 3.18  | 63.70 ± 3.69  | **66.16 ± 4.89**              |
| hopper-m-e      | 104.08 ± 5.81 | 98.99 ± 11.27 | **107.09 ± 2.69**             |
| walker-m-r      | 84.11 ± 4.12  | **90.67 ± 1.56**  | 86.02 ± 8.98              |
| walker-m        | 84.58 ± 1.92  | **85.43 ± 1.14**  | 85.42 ± 1.30              |
| walker-m-e      | 110.23 ± 0.37 | 109.95 ± 0.32 | **110.67 ± 0.89**             |
| Average         | 76.92 ± 2.66  | 78.64 ± 2.38  | **79.88 ± 3.35**              |

We also provide a general recipe for selecting hyperparameters on new tasks. While setting $(\mu=0.25, \alpha=1.1$) generally works, we observe that for low-quality datasets, increasing $\mu$ and $\alpha$ leads to further improvements by promoting exploration, as shown in the tables below.



Table2. D4RL normalized score on medium quality locomotion environments with fixed $\alpha$ and $\mu$. The experiments are conducted with TD3BC.
| Env           | None         | SynthER      | GTA($\mu = 0.5, \alpha=1.3$) | GTA($\mu = 0.75, \alpha=1.3$) |
| ------------- | ------------ | ------------ | ------------------------------ | ------------------------------- |
| halfcheetah-m | 48.42 ± 0.62 | 49.16 ± 0.39 | **57.92 ± 0.48**                   | 57.85 ± 0.27                    |
| hopper-m      | 61.04 ± 3.18 | 63.70 ± 3.69 | **68.46 ± 1.32**                   | 61.58 ± 5.00                    |
| walker-m      | 84.58 ± 1.92 | 85.43 ± 1.14 | **88.38 ± 2.70**                   | 87.14 ± 1.73                    |
| Average       | 64.68     | 66.10      | **71.59 ± 0.45**                   | 68.42 ± 2.22                    |                |


Table3. D4RL normalized score on medium-replay quality locomotion environments with fixed $\alpha$ and $\mu$. The experiments are conducted with TD3BC.
| Env | None| SynthER | GTA($\mu = 0.25, \alpha=1.1$) |
| - | - | - | - |
| halfcheetah-m-r | 44.64 ± 0.71| 45.57 ± 0.34| **46.48 ± 0.39**|
| halfcheetah-m | 48.42 ± 0.62| 49.16 ± 0.39| **49.22 ± 0.52**|
| halfcheetah-m-e | 89.48 ± 5.50| 85.47 ± 11.35 | **94.98 ± 1.66**|
| hopper-m-r| 65.69 ± 24.41 | **78.81 ± 15.80** | 72.86 ± 28.42 |
| hopper-m| 61.04 ± 3.18| 63.70 ± 3.69| **66.16 ± 4.89**|
| hopper-m-e| 104.08 ± 5.81 | 98.99 ± 11.27 | **107.09 ± 2.69** |
| walker-m-r| 84.11 ± 4.12| **90.67 ± 1.56**| 86.02 ± 8.98|
| walker-m| 84.58 ± 1.92| **85.43 ± 1.14**| 85.42 ± 1.30|
| walker-m-e| 110.23 ± 0.37 | 109.95 ± 0.32 | **110.67 ± 0.89** |
| Average | 76.92 ± 2.66| 78.64 ± 2.38| **79.88 ± 3.35**|


In summary, we propose a general configuration of hyperparameters in GTA that mostly enhances baseline performance and also provides a recipe for selecting hyperparameters to allow researchers to apply GTA for new tasks without extensive tuning.

---

> ### Author Response · Authors · 2024-08-12
> **Notice of correction to Table 3 in general response**
>
> We recently noticed that Table 3 was mistakenly uploaded as a duplicate of Table 1 in our general response. We sincerely apologize for any confusion this may have caused the reviewers and the area chair. We have now uploaded the correct Table 3, which shows a performance boost on medium-replay quality datasets, and we kindly ask you to review the updated table. Once again, we apologize for the inconvenience.
>
>
> Table 3. D4RL normalized score on medium-replay quality locomotion environments with fixed $\alpha$ and $\mu$. The experiments are conducted with TD3BC.
>
> | Env             | None          | SynthER       | GTA($\mu = 0.5, \alpha=1.3$) | GTA($\mu = 0.75, \alpha=1.3$) |
> | --------------- | ------------- | ------------- | ------------------------------ | ------------------------------- |
> | halfcheetah-m-r | 44.64 ± 0.71  | 45.57 ± 0.34  | 48.23 ± 5.42                   | **48.99 ± 2.16**                    |
> | hopper-m-r      | 65.69 ± 24.41 | 78.81 ± 15.80 | 77.17 ± 22.17                  | **97.26 ± 3.38**                    |
> | walker-m-r      | 84.11 ± 4.12  | 90.67 ± 1.56  | **91.12 ± 6.45**                   | 90.89 ± 3.29                    |
> | Average         | 64.81         | 71.68         | 72.17 ± 11.23                  | **79.05 ± 2.90**                    |

---

### Author Response · Authors · 2024-08-12

Dear Reviewers,

We hope this message finds you well.

We have noticed that reviewers **cNng** and **UiuM** have not yet participated in the discussion. As the author-reviewer discussion period is set to conclude in two days, we kindly wish to remind you of the opportunity to provide your valuable feedback.

Your insights and perspectives are incredibly important to us and will greatly contribute to improving our work. If you have any questions or require further information, please do not hesitate to reach out.

We sincerely appreciate your time and consideration, and we look forward to your input.

Best regards,
The Authors

---

### Decision · Program_Chairs · 2024-09-25

**Decision:**

Accept (poster)

**Comment:**

Reasons to accept:
- Offline RL and specifically data augmentation in RL is a research problem of practical significance.
- Paper clearly articulates a specific problem in data augmentation for offline RL and develops a method to address the problem.
- Empirical evaluation is appropriate (show casing the proposed benefit).

Reasons to reject:
- All concerns raised by the reviewers were addressed during paper discussions.

Summary: This paper identifies a significant problem for offline RL applications and develops a method for data augmentation that addresses it. An empirical study confirms the proposed benefit. The paper is clear and accessible to the RL audience at NeurIPS.